# Survival of Nematode Larvae *Strongyloides papillosus* and *Haemonchus contortus* under the Influence of Various Groups of Organic Compounds

Olexandra Boyko [1] and Viktor Brygadyrenko [2,*]

1   Department of Parasitology, Veterinary and Sanitary Examination,
    Dnipro State Agrarian and Economic University, Sergiy Efremov St., 25, 49000 Dnipro, Ukraine
2   Department of Zoology and Ecology, Oles Honchar Dnipro National University, Gagarin Av. 72,
    49010 Dnipro, Ukraine
*   Correspondence: brigad@ua.fm

**Abstract:** Many chemically synthesized xenobiotics can significantly inhibit the vitality of parasitic nematodes. However, there is yet too little research on the toxicity of such contaminating compounds toward nematodes. Compounds that are present in plants are able to inhibit the vitality of parasitic organisms as well. According to the results of our laboratory studies of toxicity, the following xenobiotics caused no decrease in the vitality of the larvae of *Strongyloides papillosus* and *Haemonchus contortus*: methanol, propan-2-ol, propylene glycol-1,2, octadecanol-1, 4-methyl-2-pentanol, 2-ethoxyethanol, butyl glycol, 2-pentanone, cyclopentanol, ortho-dimethylbenzene, dibutyl phthalate, succinic anhydride, 2-methylfuran, 2-methyl-5-nitroimidazole. Strong toxicity towards the nematode larvae was exerted by glutaraldehyde, 1,4-diethyl 2-methyl-3-oxobutanedioate, hexylamine, diethyl malonate, allyl acetoacetate, tert butyl carboxylic acid, butyl acrylate, 3-methyl-2-butanone, isobutyraldehyde, methyl acetoacetate, ethyl acetoacetate, ethyl pyruvate, 3-methylbutanal, cyclohexanol, cyclooctanone, phenol, pyrocatechin, resorcinol, naphthol-2, phenyl ether, piperonyl alcohol, 3-furoic acid, maleic anhydrid, 5-methylfurfural, thioacetic acid, butan-1-amine, dimethylformamide, 1-phenylethan-1-amine, 3-aminobenzoic acid. Widespread natural compounds (phytol, 3-hydroxy-2-butanone, maleic acid, oleic acid, hydroquinone, gallic acid-1-hydrate, taurine, 6-aminocaproic acid, glutamic acid, carnitine, ornithine monohydrochloride) had no negative effect on the larvae of *S. papillosus* and *H. contortus*. A powerful decrease in the vitality of nematode larvae was produced by 3,7-dimethyl-6-octenoic acid, isovaleric acid, glycolic acid, 2-oxopentanedioic acid, 2-methylbutanoic acid, anisole, 4-hydroxy-3-methoxybenzyl alcohol, furfuryl alcohol. The results of our studies allow us to consider 28 of the 62 compounds we studied as promising for further research on anti-nematode activity in manufacturing conditions.

**Keywords:** mortality of nematode larvae; Trichostrongylidae; Strongyloididae; anti-nematode activity; industrial contamination; combating parasites





## 1. Introduction

Various groups of organic compounds are abundant in nature and can be active against the development of parasitic nematodes during their migration in the soil and on plants. Nematodes should have become adapted to many of these compounds over millions of years of their evolution [1–3]. Xenobiotics are compounds that were absent in the natural ecosystems prior to the impact of man and which have begun to actively contaminate ecosystems in the last few centuries [4]. They include thousands of various compounds used in human and veterinary medicine, food, chemical, paint, varnish industries, household, construction, etc. Such compounds can locally contaminate natural ecosystems, decreasing the vitality of nematode larvae in small areas of waste accumulation [5–7]. Many of those compounds concentrate around urban agglomerations: they end up in landfills of solid

municipal wastes after traveling with wastewater from light and chemical industries, after being dumped as food wastes, and discharged from sewers [8,9].

Therefore, it is expected that thousands of various organic contaminants (mainly xenobiotics) should have different effects on soil nematodes. Our previous studies determined that their toxicity to nematode larvae varies broadly: $LC_{50}$ ranges from fractions of a milligram per liter to several grams per liter [10,11]. Many of those compounds have no effect on the vitality of nematodes, despite the fact that those organisms most likely had never encountered them over millions of years of their evolution [6].

Nematodes perceive the environment as gradients of concentrations of chemical signals that either attract or repel them. At high concentrations, repellents at first repel or poison them, later causing their death [12]. Larvae of parasitic nematodes (for example, species of *Haemonchus*, *Trichostrongylus*, and *Oesophagostomum genera*) spend months traveling in the upper soil horizons, ascending plants at the height of 10–50 cm above the soil surface, waiting for ruminants to eat these plants [13]. Together with rainwater, some larvae become introduced into the shallow water of water bodies (rivers, lakes, ponds) and end up in the organisms of ruminants after they drink infested water. During those life stages, nematodes are most susceptible to the influence of chemical signals—gradients of concentrations of volatile compounds generated by plants, products of life of animals, and also chemically synthesized xenobiotics contaminating the environment [14,15].

The survival of nematode larvae in solutions of many compounds that are poisonous to humans is related to the low permeability of their multi-layered cuticle and also to the anaerobic metabolism of those worms [16,17]. For example, at the first and the second stages of development, nematodes of the *Strongyloides* genus feed on various types of organic remains in soil, and, therefore, they are less tolerant to the toxic impact [18,19]. At the third larval age, these nematodes eat almost nothing (which is associated with the search for and invasion of the organism of a vertebrate host), and therefore they survive in ten-fold more concentrated solutions of compounds that are poisonous to nematodes.

People often conclude that compounds are toxic if their concentrations cause the death of 50% of laboratory vertebrate animals (*Rattus* Fischer, 1803, *Mus* Linnaeus, 1766, etc.). However, dozens of thousands of compounds, which are chemically synthesized and distributed in the conditions of urban agglomerations, are toxic to invertebrates at different doses than those established for vertebrates. Only over recent years has deep research begun into how chemically synthesized organic compounds are impacting invertebrates [20,21].

Studying contaminations that had been provoked by intense and often uncontrolled use of pesticides in agrocenoses has caused the emergence of a new direction in combat against agricultural pests: the development of biopesticides that contain those compounds. Ntalli and Caboni [22] determined the nematocidal properties of thymol in 25–250 mg/kg doses and also its activity when introduced to the soil (0, 50, 100, and 150 mg/kg) combined with benzaldehyde.

Many organic compounds occur in the essential oils of plants. Kang et al. [23] confirmed the nematocidal actions of essential oil constituents against pine nematode *Bursaphelenchus xylophilus* (Steiner and Buhrer, 1934). They evaluated the inhibiting properties of 97 compounds of essential oils (49 monoterpenes, 17 phenylpropenes, 16 sesquiterpenes, and 15 sulfides) towards the activity of the acetylcholinesterase enzyme of *B. xylophilus*.

Oka et al. [24] evaluated in vitro nematocidal activity of essential oils from 27 spicy-aromatic plants. According to their studies, essential oils from *Carum carvi* L., *Foeniculum vulgare* Mill., *Mentha rotundifolia* (L.), and *M. spicata* L. had notable nematocidal properties against root-knot nematode *M. javanica* (Treub, 1885; Chitwood, 1949). They were also able to inhibit the emergence of those nematodes from eggs. Oka [25] reported nematocidal actions towards *M. javanica* demonstrated by such constituents of essential oils as trans-cinnamaldehyde, 2-furaldehyde, benzaldehyde, and p-anisaldehyde. In our earlier articles [26,27], we reported the nematocidal properties of those compounds against nematode larvae that are parasites of ruminants.

The objective of this article was to evaluate the survival of the nematodes—wide-spread parasites of ruminants and humans (*Strongyloides papillosus* (Wedl, 1856) and *Haemonchus contortus* (Rudolphi, 1803)) in various concentrations of aqueous solutions of organic compounds that are broadly used in households, the food industry, and construction.

## 2. Materials and Methods

In the experiment, we used *Capra aegagrus hircus* goat feces (Linnaeus, 1758), infected naturally by *S. papillosus* and *H. contortus* in the territory of the Clinical Diagnostic Center of the Dnipro State Agrarian-Economic University (Dnipropetrovsk Oblast, Ukraine, coordinates: 48.421341° N, 35.051363° E). Using the generally accepted parasitological method—copro-helminth ovoscopic McMaster technique [28]—we isolated eggs of nematodes of the Rhabditida [29] order from the animals. Through 10-day cultivation at the temperature of 18–22 °C, and also using the Baermann test, we obtained free-living larvae of various ages—$L_1$, $L_2$, and $L_3$ [28]. According to their morphological specifics, we determined the following species of nematodes: *S. papillosus* of the Strongylida order and *H. contortus* of the Rhabditida order [30,31]. In the experiment, we used a mixture of different-age larvae of *Strongyloides papillosus*, and separate studies were carried out on third-age *Haemonchus contortus* larvae. When we were identifying the species, we took into account body length, total maximum body width, length of tail end, length of the esophagus, and also specifics of its structure (filiform or rhabditiform with the bulbous), length of the intestine, and specifics of its structure as well (presence or absence of notable intestinal cells, their number, form, arrangement). The experiments were carried out on non-invasive larvae (first–second stages of the development) of *S. papillosus* and also invasive larvae (third stages of the development) of *S. papillosus* and *H. contortus*.

The experimental nematode larvae, obtained using the Baermann method, were centrifuged in water for 4 min at 1500 rpm. Then, the supernatant was removed, and the sediment with larvae was uniformly distributed in 1.5 mL plastic test tubes. In the experiment, in five repetitions, we used 18–22 larvae of *S. papillosus* in each test tube (about 29,200 overall), 12–14 third-age larvae of *S. papillosus* (about 19,400 specimens), and 16–17 third-age larvae of *H. contortus* (about 2500 specimens): for 64 compounds, we tested four concentrations and the control in five repetitions of each variant of the experiment. In the experiment, we used 1%, 0.1%, 0.01%, and 0.001% solutions of the organic compounds. The larvae were exposed to those compounds for 24 h at a temperature of 22 °C. After the experiment, we counted live and dead (immobile specimens with deformed intestinal cells) nematodes.

The larvae were subjected to 62 organic compounds (Table 1) in five repetitions for each of the variants of the experiment and also in the control.

The statistical analysis of the results was performed through a set of Statistica 8.0 (StatSoft Inc., Tulsa, OK, USA). The tables present mean value (x) ± standard deviation (SD). We used the Tukey test for each of the compounds to calculate the significance of differences in the effects of various concentrations on the nematodes.

**Table 1.** Brief characteristics of the organic compounds used in the laboratory experiment.

| Name, CAS number | Formula | Use |
|---|---|---|
| Methanol CAS 67-56-1 | $CH_3OH$ | It is used in the food industry, including in some food products and beverages in low concentrations. It is a component of plastics, paints, automobile parts, and construction materials. This compound is used to fill up cars, ships, fuel cells, pots, and kitchen stoves. |
| Propan-2-ol CAS 67-63-0 | $C_3H_8O_1$ | It is used to obtain acetone as a solvent of fats, natural and synthetic resins, and nitro varnishes; also to extract alkaloids, proteins, chlorophyll, to prepare liquid soap, complex ethers, etc. In many cosmetic products and pharmaceutical drugs, this compound substitutes ethyl alcohol. Also, it is used to improve fuel quality. In households, propan-2-ol can be used to remove unpleasant smells in footwear. |

**Table 1.** *Cont.*

| Name, CAS number | Formula | Use |
|---|---|---|
| Propylene glycol-1,2 CAS 623-84-7 | $C_3H_8O_2$ | It is used in the food, pharmaceutical, and cosmetic industries. It has bactericidal properties. It is used to sterilize air, and to prepare medications (anti-inflammatory and bactericidal drugs). It is a compound in drugs for wound healing in cases of deep thermal or chemical burns, drugs that are used in veterinary medicine. In cosmetics, it is used to manufacture shampoos, emulsions, pastes, creams, lipsticks, and other preparations. |
| Octadecanol-1 (stearyl alcohol) CAS 112-92-5 | $C_{18}H_{38}O$ | It is used in lubricants, resins, perfumes, and cosmetics; manufacturing of shampoos, hair conditioners, and as an emollient additive in ointments |
| Phytol CAS 150-86-7 | $C_{20}H_{40}O$ | It is used in perfumery and cosmetics, in the making of shampoos, toilet soap, household cleaners, and detergents |
| 4-Methyl-2-pentanol CAS 108-11-2 | $C_6H_{14}O$ | In the food industry, it is used as a flavoring. It is included in glues, agents for removing glues, paints, wood stains, and similar goods. It is broadly used in the automobile industry and is also utilized in perfumery. |
| 2-Ethoxyethanol CAS 110-80-5 | $C_4H_{10}O_2$ | It is used as a solvent. In the semiconductor industry, the compound is utilized as a component of varnishes and paints. It is used for making agents for the removal of varnishes, printing ink, wood stains, and epoxy resin. |
| Butyl glycol CAS 111-76-2 | $C_6H_{14}O_2$ | In the paint and varnish industry, it is used as a solvent in paints, surface coatings, and inks. It is present in spray varnishes, agents for the removal of varnishes, paints, liquid soap, de-fattening agents, leather protectors, cleaning agents, printing pastes, emulsions, cosmetics, and herbicides. |
| Glutaraldehyde CAS 111-30-8 | $C_5H_8O_2$ | It is used as a tanning agent in the tanning industry and the production of fabrics. In microscopy, it is used for the fixation of the tissues of animals in histochemical and histological assays, especially in electronic microscopy. |
| 1,4-Diethyl 2-methyl-3-oxo-butanedioate (Diethyl 2-me-thyl-3-oxosuccinate, Diethyl oxalpropionate) CAS 759-65-9 | $C_9H_{14}O_5$ | Component for chemical synthesis. Seriously irritates skin and eyes. |
| 3,7-Dimethyl-6-octenoic acid (citronellic acid) CAS 502-47-6 | $C_{10}H_{18}O_2$ | It is a monoterpenoid formed during the oxidation of citronellal. It occurs in oil distilled using water vapor from wood and bark of coniferous trees *Callitris columellaris*, *C. glaucophylla*, and *C. intratropica*. It is found in many species of plants (*Daphne odora*, *Eucalyptus camaldulensis*, *E. exserta*, *E. tereticornis*, *E. oviformis*, *E. blakelyi*, *Pelargonium graveolens*, *P. vitifolium*, *Daphne papyracea*, *Corymbia citriodora*, *Citrus hystrix*). It is used as a flavoring for food products. |
| 3-Hydroxy-2-butanone (acetoin) CAS 513-86-0 | $C_4H_8O_2$ | The compound provides butter with a characteristic taste. Producers of hydrogenated oils usually add acetoin into the final product as an artificial buttery flavoring. It is present in apples, yogurt, asparagus, blackcurrants, blackberries, wheat, broccoli, Brussels sprouts, melons, and maple syrup. It is used as a food flavoring in bakeries. Also, it is used in liquids for electronic cigarettes, providing them with a buttery or caramel taste. |
| Hexylamine CAS 111-26-2 | $C_6H_{15}N$ | It is broadly used as a flavoring. It is added to baked foods, dry breakfast foods, cheeses, spices, fats and oils, fish products, frozen dairy products (ice cream), sauces, meat products, and ready-to-eat snacks. The compound is used for the synthesis of other chemical compounds. It is included in mixtures for vaping and for flavoring tobacco products. This compound is toxic when contacting the skin and mucous membranes and when introduced into the intestine. |
| Isovaleric acid CAS 503-74-2 | $C_5H_{10}O_2$ | In the food and perfume industries, it is used as a flavoring and fragrance because of its fruity aroma. It is used to prepare sedative medications (including validol, carvacrol, valocordin, and others). |

**Table 1.** *Cont.*

| Name, CAS number | Formula | Use |
|---|---|---|
| Glycolic acid CAS 79-14-1 | $C_2H_4O_3$ | In medicine, it is used to treat acne, aging skin, dark spots on the skin of the face, and to treat scars from acne. Also, this compound can be used in cases of skin stretches. |
| Maleic acid CAS 110-16-7 | $C_4H_4O_4$ | It is used in the manufacturing of pharmaceutic preparations. |
| Diethyl malonate CAS 105-53-3 | $C_7H_{12}O_4$ | It is used in perfumery and also for synthesis of other compounds such as barbiturates, synthetic flavorings, and vitamins $B_1$ and $B_6$. |
| 2-Oxopentanedioic acid (2-ketoglutaric acid, α-ketoglutarate) CAS 328-50-7 | $C_5H_6O_5$ | This compound is one of the most important metabolites in the Krebs cycle. A high concentration of α-ketoglutarate in an organism is associated with an increase in the life spans of nematodes and mice. It promotes the differentiation of naïve CD4+ T-cells in TH1, inhibiting their differentiation into anti-inflammatory Treg-cells. |
| 2-Methylbutanoic acid CAS 116-53-0 | $C_5H_{10}O_2$ | It is used as a food flavoring, cleaner, and air freshener. It is present in cacao beans and many fruits (apples, apricots), roots of *Valeriana officinalis*. Raceme-like 2-methylbutanoic acid has a pungent quark smell. (S)-2-methylbutanoic acid has a pleasant sweet fruit aroma, whereas (R)-2-methylbutanoic acid has an unpleasant cheese-like odor of sweat. It is present in the sweat of people and many mammals. |
| Allyl acetoacetate CAS 1118-84-9 | $C_7H_{10}O_3$ | It is used for the synthesis of pharmaceutical drugs. It is toxic when contacting the skin and when swallowed. |
| Tert-butyl carboxylic acid (pivalic acid) CAS 75-98-9 | $C_5H_{10}O_2$ | It is used in the production of polyvinyl ethers (vinyl pivalate) and pharmaceutical drugs. Annually, several million kilograms of pivalic acid are produced globally. |
| Butyl acrylate CAS 141-32-2 | $C_7H_{12}O_2$ | It is used in paints, sealants, coatings, adhesives, fuel, textiles, plastics, and caulk. |
| 3-Methyl-2-butanone CAS 563-80-4 | $C_5H_{10}O$ | Food improvement agents. It is used as a flavoring. It is used in perfumery. It can be used as a solvent. |
| Isobutyraldehyde (2-methylpropanal) CAS 78-84-2 | $C_4H_8O$ | Isobutyraldehyde (2-methylpropanal) is obtained in massive amounts by hydroformylation of propylene. Annually, several million tonnes are produced. Its odor is described as such of wet straw. |
| Methyl acetoacetate CAS 105-45-3 | $C_5H_8O_3$ | It is used for manufacturing paints for houses, as a flavoring, and as a component of aromatic mixtures. |
| Ethyl acetoacetate CAS 141-97-9 | $C_6H_{10}O_3$ | It is used in the chemical synthesis of various compounds and as a flavoring in the food industry. It is included in cleaners, agents for households (air fresheners), and scented candles. The compound is used for manufacturing pesticides, including insecticides. |
| Ethyl pyruvate CAS 617-35-6 | $C_5H_8O_3$ | It is used as a flavoring in the food industry and perfumery. |
| 3-Methylbutanal (isovaleraldehyde) CAS 590-86-3 | $C_5H_{10}O$ | It is used in the food industry as a flavoring. It is used in the chemical synthesis of pesticides and pharmaceutical preparations. |
| Oleic acid CAS 112-80-1 | $C_{18}H_{34}O_2$ | It is used in cosmetology and is included in agents of chemical wave, lipsticks, and agents for skin and hair care. It is used to make soft kinds of soap. |
| 2-Pentanone CAS 107-87-9 | $C_5H_{10}O$ | It is used as a solvent of varnishes and surface coatings, for cleaning and de-fattening surfaces, and as a flavoring in the food industry. |
| Cyclopentanol CAS 96-41-3 | $C_5H_{10}O$ | It is used as a flavoring in the food industry, as a solvent when making perfume-pharmaceutical preparations, and also for organic synthesis. |
| Cyclohexanol CAS 108-93-0 | $C_6H_{12}O$ | It is used for manufacturing cleaners and agents for household use. It is also used as a de-fattening agent. It is used as a hygiene agent in the production of goods for children. The compound is also used as a flavoring. |

**Table 1.** *Cont.*

| Name, CAS number | Formula | Use |
|---|---|---|
| Cyclooctanone CAS 502-49-8 | $C_8H_{14}O$ | It is used for the treatment of cardiovascular diseases that are related to an abnormally high level of aldosterone. It is used for the synthesis of 14-membered lactones. |
| Phenol (carbolic acid) CAS 108-95-2 | $C_6H_6O$ | It is used as a precursor for the synthesis of various plastics. This compound is a precursor to some drugs and many pesticides. It is used as an antiseptic. It is a component of industrial solvents for the removal of paint. In cosmetology, it is used as a component of creams, hair dyes, and preparations for skin lightening. |
| Pyrocatechin CAS 120-80-9 | $C_6H_6O_2$ | In photography, the compound is used as a developing agent. It is used in the production of colorings and medicinal compounds. |
| Resorcinol CAS 108-46-3 | $C_6H_6O_2$ | In medicine, it is used as an antiseptic and disinfecting agent. It has keratolytic and fungicidal effects. |
| Hydroquinone CAS 123-31-9 | $C_6H_6O_2$ | It is used as a developing agent in photography, an antioxidant in the chemical industry, and a reagent for the identification of wolfram, gold, and cesium in analytical chemistry. In the food industry, it is used as an antioxidant. It is also used in cosmetology and cosmetic medicine. |
| Naphthol-2 CAS 135-19-3 | $C_{10}H_8O$ | It is used to obtain sudan, acidic orange (azo dyes). In pharmaceutics, its methyl and ethyl ethers are used to prepare drugs. |
| Anisole CAS 100-66-3 | $C_7H_8O$ | It is used as a food flavoring. |
| Diphenyl ether (phenyl ether) CAS 101-84-8 | $C_{12}H_{10}O$ | It is used as a cleaner, agent for household appliances, air freshener, and fragrance. It is a component for preparations to combat microorganisms on solid surfaces or disinfecting goods that are washed. In perfumery, it is used for making soap and soap perfumes. |
| ortho-dimethylbenzene (o-Xylene) CAS 95-47-6 | $C_8H_{10}$ | It is used to obtain phthalic anhydride and phthalic acid. The compound is used as a solvent for varnishes, paints, and putty. It is a component of some liquids for the removal of oil and fat from solid surfaces. |
| Piperonyl alcohol CAS 495-76-1 | $C_8H_8O_3$ | It is used as a food flavoring. In air, it breaks down under the influence of hydroxyl radicals (half-life period is 7 h). |
| 4-Hydroxy-3-methoxybenzyl alcohol (vanillyl alcohol) CAS 498-00-0 | $C_8H_{10}O_3$ | It is used as a food flavoring. It irritates human skin and eyes. |
| Gallic acid-1-hydrate CAS 5995-86-8 | $C_7H_8O_6$ | It is used for the synthesis of pyrogallic acid, drugs, ink, and colorings. It is used for the detection of free inorganic acid, dihydroxyacetone, and alkaloids. It is used as a developing agent and food preservative to prepare propyl gallate. |
| Dibutyl phthalate CAS 84-74-2 | $C_{16}H_{22}O_4$ | It is a broadly used plastifier and is used to prepare many engineering plastics, including polyvinyl chloride. |
| 3-Furoic acid CAS 488-93-7 | $C_5H_4O_3$ | Reagent for chemical synthesis. It seriously irritates skin, eyes, and airways. |
| Succinic anhydride CAS 108-30-5 | $C_4H_4O_3$ | The compound is used in the chemical industry, cosmetology, and pharmaceuticals; as an adhesive agent and solidifier for epoxy resin. It is also utilized in paper manufacturing. |
| Maleic anhydride CAS 108-31-6 | $C_4H_2O_3$ | It is used in the production of unsaturated polyester resins, thermoplastic polyurethanes, and elastane fibers, for the synthesis of agricultural pesticides; used in the food industry to manufacture food additives (fumaric, succinic, and malic acids). |
| 5-Methylfurfural CAS 620-02-0 | $C_6H_6O_2$ | In the food industry, it is used as a flavoring. It is a component of cleaners and air fresheners. It is also used for the manufacture of aromatic candles. |
| 2-Methylfuran CAS 534-22-5 | $C_5H_6O$ | In the food industry, it is used as a flavoring and adjuvant. It seriously irritates the eyes and is dangerous for people if swallowed. |

**Table 1.** *Cont.*

| Name, CAS number | Formula | Use |
|---|---|---|
| Furfuryl alcohol CAS 98-00-0 | $C_5H_6O_2$ | The compound is used as a food additive. It accumulates in biomass wastes as a result of microbiological decomposition (corncobs, press cakes of sugar cane, and others). Furfuryl alcohol is used as fuel in rocket technologies. It is a constituent of glue and agents for glue removal and sealants, agents used for long-scale coating and protection of wooden surfaces, and agents for paint removal. |
| Thioacetic acid CAS 507-09-5 | $C_2H_4OS$ | It is used in organic synthesis and in cosmetology. |
| Taurine CAS 107-35-7 | $C_2H_7NO_3S$ | It is used as a food additive or medicinal agent and an ingredient in energy beverages. In medicine, it is used for the treatment of diabetes and other diseases. The compound is broadly used in the sports nutrition industry and is used as a medication and biological additive. In cosmetology, it is used as an anti-aging cosmetic. |
| Butan-1-amine CAS 109-73-9 | $C_4H_{11}N$ | It is used as an intermediate in the synthesis of dyes, drugs, rubber additives, emulsifiers, tanning agents, and insecticides; also used as a vulcanizing accelerator for rubber and as a curing agent for polymers. |
| 6-Aminocaproic acid CAS 60-32-2 | $C_6H_{13}NO_2$ | In medicine, this acid was approved by the FDA of the USA for the treatment of severe hemorrhages associated with increased fibrinolytic activity. It is used to stop bleeding. |
| Dimethylformamide CAS 68-12-2 | $C_3H_7NO$ | It is used as a solvent in the production of polyacrylonitrile fiber (nitron) and other polymers and as a solvent of colorings for dying leather, paper, wood, and viscose. |
| Glutamic acid CAS 56-86-0 | $C_5H_9NO_4$ | It is used as a food additive (E620) and in pharmaceutics. As a flavor enhancer in the food industry, there is a broad application of salts of this compound: monosodium glutamate (E621), potassium (E622), ammonium (E624), magnesium (E625), calcium D Glutamate (E623). |
| L-Carnitine CAS 541-15-1 | $C_7H_{15}NO_3$ | In medicine, and also fitness training and bodybuilding, it is used for the correction of metabolic processes. It has anabolic, antihypoxic, and antithyroid effects, activates fat metabolism, stimulates regeneration, and enhances appetite. |
| Ornithine hydrochloride CAS 3184-13-2 | $C_5H_{12}N_2O_2$ | Ornithine is a non-proteinogenic amino acid that plays a role in the urea cycle. Ornithine is not an amino acid coded for by DNA, that is, not proteinogenic. However, in mammalian non-hepatic tissues, the main use of the urea cycle is in arginine biosynthesis, so ornithine is quite important as an intermediate in metabolic processes. |
| 1-Phenylethan-1-amine CAS 618-36-0 | $C_8H_{11}N$ | In the food industry, it is used as an emulsifier for synthesis and also as a resolving agent. |
| 3-Aminobenzoic acid CAS 99-05-8 | $C_7H_8ClNO_2$ | It is used for the synthesis of azo dyes to provide cellulose fibers with red, yellow, and brown colors. It irritates skin, eyes, and airways. |
| 2-Methyl-5-nitroimidazole CAS 88054-22-2 | $C_4H_5N_3O_2$ | It is an intermediate product in the synthesis reaction of tinidazole—the most important medicinal agent that is effective against amoebiasis, trichomoniasis, giardiasis, acute ulcerative gingivitis, and post-operational anaerobic infections. It is used for treating almost all protozoan infections. |

The data are generalized based on Svobodová et al. [32], with additions of information from other sources (https://pubchem.ncbi.nlm.nih.gov accessed on 10 January 2023, www.wikidata.org accessed on 10 January 2023).

## 3. Results

The greatest negative impacts on the nematode larvae were exerted by glutaraldehyde, thioacetic acid, 3-furoic acid, diethyl malonate, 2-oxopentanedioic acid, butan-1-amine, isovaleric acid, ethyl acetoacetate, phenol and naphthol-2. Twenty-four-hour exposure to 1% solutions of those compounds killed 100% of the *S. papillosus* larvae of all the development stages and also the *H. contortus* larvae of the third (invasive) stage (Tables 2–4).

**Table 2.** Mortality of larvae of *S. papillosus* and *H. contortus* (%) during 24 h laboratory experiment under the influence of acyclic organic compounds (x ± SD, each experiment was repeated five times).

| Compound | Nematode Species | Mortality of Nematode Larvae in Control, % | Mortality of Nematode Larvae in 1.0% Solution, % | Mortality of Nematode Larvae in 0.1% Solution, % | Mortality of Nematode Larvae in 0.01% Solution, % | Mortality of Nematode Larvae in 0.001% Solution, % | $LC_{50}$, % * |
|---|---|---|---|---|---|---|---|
| Methanol | $L_{1-2}$ of *S. papillosus* | 0.0 ± 0.0 [a] | 8.0 ± 12.0 [a] | 0.0 ± 0.0 [a] | 0.0 ± 0.0 [a] | 0.0 ± 0.0 [a] | – |
|  | $L_3$ of *S. papillosus* | 0.0 ± 0.0 [a] | 0.0 ± 0.0 [a] | 0.0 ± 0.0 [a] | 0.0 ± 0.0 [a] | 0.0 ± 0.0 [a] | – |
|  | $L_3$ of *H. contortus* | 0.0 ± 0.0 [a] | 0.0 ± 0.0 [a] | 0.0 ± 0.0 [a] | 0.0 ± 0.0 [a] | 0.0 ± 0.0 [a] | – |
| Propan-2-ol | $L_{1-2}$ of *S. papillosus* | 0.0 ± 0.0 [a] | 0.0 ± 0.0 [a] | 0.0 ± 0.0 [a] | 0.0 ± 0.0 [a] | 0.0 ± 0.0 [a] | – |
|  | $L_3$ of *S. papillosus* | 0.0 ± 0.0 [a] | 0.0 ± 0.0 [a] | 0.0 ± 0.0 [a] | 0.0 ± 0.0 [a] | 0.0 ± 0.0 [a] | – |
|  | $L_3$ of *H. contortus* | 0.0 ± 0.0 [a] | 0.0 ± 0.0 [a] | 0.0 ± 0.0 [a] | 0.0 ± 0.0 [a] | 0.0 ± 0.0 [a] | – |
| Propylene glycol-1,2 | $L_{1-2}$ of *S. papillosus* | 0.0 ± 0.0 [a] | 25.2 ± 12.1 [b] | 0.0 ± 0.0 [a] | 0.0 ± 0.0 [a] | 0.0 ± 0.0 [a] | – |
|  | $L_3$ of *S. papillosus* | 0.0 ± 0.0 [a] | 10.0 ± 5.8 [b] | 0.0 ± 0.0 [a] | 0.0 ± 0.0 [a] | 0.0 ± 0.0 [a] | – |
|  | $L_3$ of *H. contortus* | 0.0 ± 0.0 [a] | 0.0 ± 0.0 [a] | 0.0 ± 0.0 [a] | 0.0 ± 0.0 [a] | 0.0 ± 0.0 [a] | – |
| Octadecanol-1 | $L_{1-2}$ of *S. papillosus* | 0.0 ± 0.0 [a] | 0.0 ± 0.0 [a] | 0.0 ± 0.0 [a] | 0.0 ± 0.0 [a] | 0.0 ± 0.0 [a] | – |
|  | $L_3$ of *S. papillosus* | 0.0 ± 0.0 [a] | 0.0 ± 0.0 [a] | 0.0 ± 0.0 [a] | 0.0 ± 0.0 [a] | 0.0 ± 0.0 [a] | – |
|  | $L_3$ of *H. contortus* | 0.0 ± 0.0 [a] | 0.0 ± 0.0 [a] | 0.0 ± 0.0 [a] | 0.0 ± 0.0 [a] | 0.0 ± 0.0 [a] | – |
| Phytol | $L_{1-2}$ of *S. papillosus* | 0.0 ± 0.0 [a] | 5.0 ± 8.3 [a] | 0.0 ± 0.0 [a] | 0.0 ± 0.0 [a] | 0.0 ± 0.0 [a] | – |
|  | $L_3$ of *S. papillosus* | 0.0 ± 0.0 [a] | 0.0 ± 0.0 [a] | 0.0 ± 0.0 [a] | 0.0 ± 0.0 [a] | 0.0 ± 0.0 [a] | – |
|  | $L_3$ of *H. contortus* | 0.0 ± 0.0 [a] | 0.0 ± 0.0 [a] | 0.0 ± 0.0 [a] | 0.0 ± 0.0 [a] | 0.0 ± 0.0 [a] | – |
| 4-Methyl-2-pentanol | $L_{1-2}$ of *S. papillosus* | 0.0 ± 0.0 [a] | 10.0 ± 12.5 [a] | 0.0 ± 0.0 [a] | 0.0 ± 0.0 [a] | 0.0 ± 0.0 [a] | – |
|  | $L_3$ of *S. papillosus* | 0.0 ± 0.0 [a] | 0.0 ± 0.0 [a] | 0.0 ± 0.0 [a] | 0.0 ± 0.0 [a] | 0.0 ± 0.0 [a] | – |
|  | $L_3$ of *H. contortus* | 0.0 ± 0.0 [a] | 0.0 ± 0.0 [a] | 0.0 ± 0.0 [a] | 0.0 ± 0.0 [a] | 0.0 ± 0.0 [a] | – |
| 2-Ethoxyethanol | $L_{1-2}$ of *S. papillosus* | 0.0 ± 0.0 [a] | 16.4 ± 4.0 [b] | 0.0 ± 0.0 [a] | 0.0 ± 0.0 [a] | 0.0 ± 0.0 [a] | – |
|  | $L_3$ of *S. papillosus* | 0.0 ± 0.0 [a] | 0.0 ± 0.0 [a] | 0.0 ± 0.0 [a] | 0.0 ± 0.0 [a] | 0.0 ± 0.0 [a] | – |
|  | $L_3$ of *H. contortus* | 0.0 ± 0.0 [a] | 0.0 ± 0.0 [a] | 0.0 ± 0.0 [a] | 0.0 ± 0.0 [a] | 0.0 ± 0.0 [a] | – |
| Butyl glycol | $L_{1-2}$ of *S. papillosus* | 0.0 ± 0.0 [a] | 30.0 ± 10.0 [b] | 0.0 ± 0.0 [a] | 0.0 ± 0.0 [a] | 0.0 ± 0.0 [a] | – |
|  | $L_3$ of *S. papillosus* | 0.0 ± 0.0 [a] | 0.0 ± 0.0 [a] | 0.0 ± 0.0 [a] | 0.0 ± 0.0 [a] | 0.0 ± 0.0 [a] | – |
|  | $L_3$ of *H. contortus* | 0.0 ± 0.0 [a] | 0.0 ± 0.0 [a] | 0.0 ± 0.0 [a] | 0.0 ± 0.0 [a] | 0.0 ± 0.0 [a] | – |
| Glutaraldehyde | $L^{1-2}$ of *S. papillosus* | 0.0 ± 0.0 [a] | 100.0 ± 0.0 [b] | 65.8 ± 3.2 [c] | 0.0 ± 0.0 [a] | 0.0 ± 0.0 [a] | 0.0784 ± 0.0033 |
|  | $L_3$ of *S. papillosus* | 0.0 ± 0.0 [a] | 100.0 ± 0.0 [b] | 48.0 ± 9.8 [c] | 0.0 ± 0.0 [a] | 0.0 ± 0.0 [a] | 0.1346 ± 0.1691 |
|  | $L_3$ of *H. contortus* | 0.0 ± 0.0 [a] | 100.0 ± 0.0 [b] | 6.9 ± 9.6 [c] | 0.0 ± 0.0 [a] | 0.0 ± 0.0 [a] | 0.5166 ± 0.0504 |
| 1,4-Diethyl-2-methyl-3-oxobutanedioate | $L_{1-2}$ of *S. papillosus* | 0.0 ± 0.0 [a] | 100.0 ± 0.0 [b] | 11.2 ± 11.0 [c] | 0.0 ± 0.0 [a] | 0.0 ± 0.0 [a] | 0.4932 ± 0.0638 |
|  | $L_3$ of *S. papillosus* | 0.0 ± 0.0 [a] | 98.1 ± 2.6 [b] | 0.0 ± 0.0 [a] | 0.0 ± 0.0 [a] | 0.0 ± 0.0 [a] | 0.5587 ± 0.0122 |
|  | $L_3$ of *H. contortus* | 0.0 ± 0.0 [a] | 48.6 ± 17.0 [b] | 0.0 ± 0.0 [a] | 0.0 ± 0.0 [a] | 0.0 ± 0.0 [a] | – |
| 3,7-Dimethyl-6-octenoic acid | $L_{1-2}$ of *S. papillosus* | 0.0 ± 0.0 [a] | 100.0 ± 0.0 [b] | 100.0 ± 0.0 [b] | 1.9 ± 2.7 [a] | 0.0 ± 0.0 [a] | 0.0520 ± 0.0044 |
|  | $L_3$ of *S. papillosus* | 0.0 ± 0.0 [a] | 100.0 ± 0.0 [b] | 62.1 ± 15.8 [c] | 0.0 ± 0.0 [a] | 0.0 ± 0.0 [a] | 0.0825 ± 0.0197 |
|  | $L_3$ of *H. contortus* | 0.0 ± 0.0 [a] | 72.0 ± 25.9 [b] | 20.7 ± 21.7 [c] | 0.0 ± 0.0 [a] | 0.0 ± 0.0 [a] | 0.6140 ± 0.4256 |
| 3-Hydroxy-2-butanone | $L_{1-2}$ of *S. papillosus* | 0.0 ± 0.0 [a] | 72.8 ± 4.3 [b] | 56.7 ± 12.8 [c] | 0.0 ± 0.0 [a] | 0.0 ± 0.0 [a] | 0.0894 ± 0.0189 |
|  | $L_3$ of *S. papillosus* | 0.0 ± 0.0 [a] | 18.9 ± 5.6 [b] | 7.2 ± 2.3 [c] | 0.0 ± 0.0 [a] | 0.0 ± 0.0 [a] | – |
|  | $L_3$ of *H. contortus* | 0.0 ± 0.0 [a] | 0.0 ± 0.0 [a] | 0.0 ± 0.0 [a] | 0.0 ± 0.0 [a] | 0.0 ± 0.0 [a] | – |
| Hexylamine | $L_{1-2}$ of *S. papillosus* | 0.0 ± 0.0 [a] | 100.0 ± 0.0 [b] | 100.0 ± 0.0 [b] | 72.5 ± 2.1 [c] | 26.4 ± 13.2 [d] | 0.0056 ± 0.0016 |
|  | $L_3$ of *S. papillosus* | 0.0 ± 0.0 [a] | 100.0 ± 0.0 [b] | 100.0 ± 0.0 [b] | 27.1 ± 2.9 [c] | 20.0 ± 16.3 [c] | 0.0383 ± 0.0025 |
|  | $L_3$ of *H. contortus* | 0.0 ± 0.0 [a] | 100.0 ± 0.0 [b] | 20.0 ± 11.7 [c] | 0.0 ± 0.0 [a] | 0.0 ± 0.0 [a] | 0.4375 ± 0.0841 |

**Table 2.** *Cont.*

| Compound | Nematode Species | Mortality of Nematode Larvae in Control, % | Mortality of Nematode Larvae in 1.0% Solution, % | Mortality of Nematode Larvae in 0.1% Solution, % | Mortality of Nematode Larvae in 0.01% Solution, % | Mortality of Nematode Larvae in 0.001% Solution, % | $LC_{50}$, % * |
|---|---|---|---|---|---|---|---|
| Isovaleric acid | $L_{1-2}$ of *S. papillosus* | 0.0 ± 0.0 [a] | 100.0 ± 0.0 [b] | 100.0 ± 0.0 [b] | 12.2 ± 3.7 [c] | 0.0 ± 0.0 [a] | 0.0487 ± 0.0022 |
|  | $L_3$ of *S. papillosus* | 0.0 ± 0.0 [a] | 100.0 ± 0.0 [b] | 85.6 ± 8.5 [c] | 0.0 ± 0.0 [a] | 0.0 ± 0.0 [a] | 0.0626 ± 0.0053 |
|  | $L_3$ of *H. contortus* | 0.0 ± 0.0 [a] | 100.0 ± 0.0 [b] | 2.0 ± 2.8 [a] | 0.0 ± 0.0 [a] | 0.0 ± 0.0 [a] | 0.5408 ± 0.0131 |
| Glycolic acid | $L_{1-2}$ of *S. papillosus* | 0.0 ± 0.0 [a] | 100.0 ± 0.0 [b] | 100.0 ± 0.0 [b] | 4.5 ± 3.7 [c] | 0.0 ± 0.0 [a] | 0.0529 ± 0.0018 |
|  | $L_3$ of *S. papillosus* | 0.0 ± 0.0 [a] | 100.0 ± 0.0 [b] | 67.6 ± 8.4 [c] | 0.9 ± 1.9 [a] | 0.0 ± 0.0 [a] | 0.0763 ± 0.0091 |
|  | $L_3$ of *H. contortus* | 0.0 ± 0.0 [a] | 11.7 ± 16.2 [a] | 0.0 ± 0.0 [a] | 0.0 ± 0.0 [a] | 0.0 ± 0.0 [a] | − |
| Maleic acid | $L_{1-2}$ of *S. papillosus* | 0.0 ± 0.0 [a] | 0.0 ± 0.0 [a] | 0.0 ± 0.0 [a] | 0.0 ± 0.0 [a] | 0.0 ± 0.0 [a] | − |
|  | $L_3$ of *S. papillosus* | 0.0 ± 0.0 [a] | 0.0 ± 0.0 [a] | 0.0 ± 0.0 [a] | 0.0 ± 0.0 [a] | 0.0 ± 0.0 [a] | − |
|  | $L_3$ of *H. contortus* | 0.0 ± 0.0 [a] | 0.0 ± 0.0 [a] | 0.0 ± 0.0 [a] | 0.0 ± 0.0 [a] | 0.0 ± 0.0 [a] | − |
| Diethyl malonate | $L_{1-2}$ of *S. papillosus* | 0.0 ± 0.0 [a] | 100.0 ± 0.0 [b] | 100.0 ± 0.0 [b] | 20.4 ± 5.2 [c] | 0.0 ± 0.0 [a] | 0.0435 ± 0.0037 |
|  | $L_3$ of *S. papillosus* | 0.0 ± 0.0 [a] | 100.0 ± 0.0 [b] | 100.0 ± 0.0 [b] | 17.9 ± 4.4 [c] | 0.0 ± 0.0 [a] | 0.0452 ± 0.0029 |
|  | $L_3$ of *H. contortus* | 0.0 ± 0.0 [a] | 100.0 ± 0.0 [b] | 100.0 ± 0.0 [b] | 47.3 ± 18.3 [c] | 0.0 ± 0.0 [a] | 0.0146 ± 0.0337 |
| 2-Oxopentanedioic acid | $L_{1-2}$ of *S. papillosus* | 0.0 ± 0.0 [a] | 100.0 ± 0.0 [b] | 100.0 ± 0.0 [b] | 100.0 ± 0.0 [b] | 44.3 ± 11.7 [c] | 0.0019 ± 0.0018 |
|  | $L_3$ of *S. papillosus* | 0.0 ± 0.0 [a] | 100.0 ± 0.0 [b] | 100.0 ± 0.0 [b] | 96.7 ± 7.5 [b] | 27.7 ± 8.8 [c] | 0.0039 ± 0.0011 |
|  | $L_3$ of *H. contortus* | 0.0 ± 0.0 [a] | 100.0 ± 0.0 [b] | 100.0 ± 0.0 [b] | 36.7 ± 21.7 [c] | 0.0 ± 0.0 [a] | 0.0289 ± 0.0276 |
| 2-Methylbutanoic acid | $L_{1-2}$ of *S. papillosus* | 0.0 ± 0.0 [a] | 100.0 ± 0.0 [b] | 100.0 ± 0.0 [b] | 100.0 ± 0.0 [b] | 43.6 ± 9.7 [c] | 0.0020 ± 0.0014 |
|  | $L_3$ of *S. papillosus* | 0.0 ± 0.0 [a] | 100.0 ± 0.0 [b] | 81.3 ± 18.0 [c] | 76.7 ± 13.7 [c] | 26.4 ± 11.9 [d] | 0.0052 ± 0.0023 |
|  | $L_3$ of *H. contortus* | 0.0 ± 0.0 [a] | 40.0 ± 22.4 [b] | 4.0 ± 8.9 [c] | 0.0 ± 0.0 [a] | 0.0 ± 0.0 [a] | − |
| Allyl acetoacetate | $L_{1-2}$ of *S. papillosus* | 0.0 ± 0.0 [a] | 100.0 ± 0.0 [b] | 100.0 ± 0.0 [b] | 68.6 ± 7.3 [c] | 7.6 ± 7.0 [d] | 0.0073 ± 0.0011 |
|  | $L_3$ of *S. papillosus* | 0.0 ± 0.0 [a] | 100.0 ± 0.0 [b] | 39.8 ± 3.0 [c] | 0.0 ± 0.0 [a] | 0.0 ± 0.0 [a] | 0.2525 ± 0.0373 |
|  | $L_3$ of *H. contortus* | 0.0 ± 0.0 [a] | 77.6 ± 14.4 [b] | 0.0 ± 0.0 [a] | 0.0 ± 0.0 [a] | 0.0 ± 0.0 [a] | 0.6799 ± 0.1114 |
| Tert butyl carboxylic acid | $L_{1-2}$ of *S. papillosus* | 0.0 ± 0.0 [a] | 100.0 ± 0.0 [b] | 77.5 ± 6.5 [c] | 0.0 ± 0.0 [a] | 0.0 ± 0.0 [a] | 0.0681 ± 0.0049 |
|  | $L_3$ of *S. papillosus* | 0.0 ± 0.0 [a] | 100.0 ± 0.0 [b] | 5.4 ± 7.4 [a] | 0.0 ± 0.0 [a] | 0.0 ± 0.0 [a] | 0.5243 ± 0.0374 |
|  | $L_3$ of *H. contortus* | 0.0 ± 0.0 [a] | 83.3 ± 23.6 [b] | 0.0 ± 0.0 [a] | 0.0 ± 0.0 [a] | 0.0 ± 0.0 [a] | 0.6402 ± 0.1664 |
| Butyl acrylate | $L_{1-2}$ of *S. papillosus* | 0.0 ± 0.0 [a] | 100.0 ± 0.0 [b] | 100.0 ± 0.0 [b] | 48.8 ± 4.7 [c] | 0.0 ± 0.0 [a] | 0.0121 ± 0.0081 |
|  | $L_3$ of *S. papillosus* | 0.0 ± 0.0 [a] | 100.0 ± 0.0 [b] | 100.0 ± 0.0 [b] | 29.7 ± 5.8 [c] | 0.0 ± 0.0 [a] | 0.0360 ± 0.0053 |
|  | $L_3$ of *H. contortus* | 0.0 ± 0.0 [a] | 68.0 ± 31.8 [b] | 37.1 ± 16.0 [bc] | 20.0 ± 18.3 [c] | 0.0 ± 0.0 [a] | 0.4757 ± 0.8911 |
| 3-Methyl-2-butanone | $L_{1-2}$ of *S. papillosus* | 0.0 ± 0.0 [a] | 97.9 ± 1.9 [b] | 28.4 ± 1.2 [c] | 0.0 ± 0.0 [a] | 0.0 ± 0.0 [a] | 0.3797 ± 0.0184 |
|  | $L_3$ of *S. papillosus* | 0.0 ± 0.0 [a] | 51.5 ± 7.0 [b] | 0.0 ± 0.0 [a] | 0.0 ± 0.0 [a] | 0.0 ± 0.0 [a] | 0.9738 ± 0.1210 |
|  | $L_3$ of *H. contortus* | 0.0 ± 0.0 [a] | 0.0 ± 0.0 [a] | 0.0 ± 0.0 [a] | 0.0 ± 0.0 [a] | 0.0 ± 0.0 [a] | − |

**Table 2.** *Cont.*

| Compound | Nematode Species | Mortality of Nematode Larvae in Control, % | Mortality of Nematode Larvae in 1.0% Solution, % | Mortality of Nematode Larvae in 0.1% Solution, % | Mortality of Nematode Larvae in 0.01% Solution, % | Mortality of Nematode Larvae in 0.001% Solution, % | LC$_{50}$, % * |
|---|---|---|---|---|---|---|---|
| Isobutyraldehyde | L$_{1–2}$ of *S. papillosus* | 0.0 ± 0.0 [a] | 100.0 ± 0.0 [b] | 100.0 ± 0.0 [b] | 4.8 ± 1.7 [c] | 0.0 ± 0.0 [a] | 0.0527 ± 0.0008 |
| | L$_3$ of *S. papillosus* | 0.0 ± 0.0 [a] | 100.0 ± 0.0 [b] | 75.5 ± 11.0 [c] | 0.0 ± 0.0 [a] | 0.0 ± 0.0 [a] | 0.0696 ± 0.0089 |
| | L$_3$ of *H. contortus* | 0.0 ± 0.0 [a] | 76.3 ± 15.3 [b] | 4.0 ± 8.9 [a] | 0.0 ± 0.0 [a] | 0.0 ± 0.0 [a] | 0.6726 ± 0.1628 |
| Methyl acetoacetate | L$_{1–2}$ of *S. papillosus* | 0.0 ± 0.0 [a] | 100.0 ± 0.0 [b] | 91.4 ± 5.7 [c] | 0.0 ± 0.0 [a] | 0.0 ± 0.0 [a] | 0.0592 ± 0.0031 |
| | L$_3$ of *S. papillosus* | 0.0 ± 0.0 [a] | 100.0 ± 0.0 [b] | 16.7 ± 4.4 [c] | 0.0 ± 0.0 [a] | 0.0 ± 0.0 [a] | 0.4598 ± 0.0286 |
| | L$_3$ of *H. contortus* | 0.0 ± 0.0 [a] | 25.0 ± 14.4 [b] | 0.0 ± 0.0 [a] | 0.0 ± 0.0 [a] | 0.0 ± 0.0 [a] | – |
| Ethyl acetoacetate | L$_{1–2}$ of *S. papillosus* | 0.0 ± 0.0 [a] | 100.0 ± 0.0 [b] | 100.0 ± 0.0 [b] | 73.0 ± 9.6 [c] | 15.4 ± 10.4 [d] | 0.0064 ± 0.0016 |
| | L$_3$ of *S. papillosus* | 0.0 ± 0.0 [a] | 100.0 ± 0.0 [b] | 32.3 ± 10.9 [c] | 0.0 ± 0.0 [a] | 0.0 ± 0.0 [a] | 0.3353 ± 0.1099 |
| | L$_3$ of *H. contortus* | 0.0 ± 0.0 [a] | 100.0 ± 0.0 [b] | 9.0 ± 12.4 [a] | 0.0 ± 0.0 [a] | 0.0 ± 0.0 [a] | 0.5055 ± 0.0687 |
| Ethyl pyruvate | L$_{1–2}$ of *S. papillosus* | 0.0 ± 0.0 [a] | 100.0 ± 0.0 [b] | 44.2 ± 9.0 [c] | 26.4 ± 4.0 [d] | 0.0 ± 0.0 [a] | 0.1935 ± 0.1335 |
| | L$_3$ of *S. papillosus* | 0.0 ± 0.0 [a] | 93.3 ± 14.9 [b] | 0.0 ± 0.0 [a] | 0.0 ± 0.0 [a] | 0.0 ± 0.0 [a] | 0.5823 ± 0.0790 |
| | L$_3$ of *H. contortus* | 0.0 ± 0.0 [a] | 92.7 ± 10.1 [b] | 0.0 ± 0.0 [a] | 0.0 ± 0.0 [a] | 0.0 ± 0.0 [a] | 0.5854 ± 0.0535 |
| 3-Methylbutanal (Isovaleraldehyde) | L$_{1–2}$ of *S. papillosus* | 0.0 ± 0.0 [a] | 100.0 ± 0.0 [b] | 100.0 ± 0.0 [b] | 61.6 ± 5.7 [c] | 26.3 ± 3.9 [d] | 0.0070 ± 0.0013 |
| | L$_3$ of *S. papillosus* | 0.0 ± 0.0 [a] | 100.0 ± 0.0 [b] | 66.7 ± 8.2 [c] | 15.2 ± 10.2 [d] | 0.0 ± 0.0 [a] | 0.0708 ± 0.0155 |
| | L$_3$ of *H. contortus* | 0.0 ± 0.0 [a] | 49.3 ± 5.7 [b] | 33.1 ± 6.6 [c] | 17.9 ± 5.0 [d] | 0.0 ± 0.0 [a] | – |
| Oleic acid | L$_{1–2}$ of *S. papillosus* | 0.0 ± 0.0 [a] | 23.5 ± 12.5 [b] | 0.0 ± 0.0 [a] | 0.0 ± 0.0 [a] | 0.0 ± 0.0 [a] | – |
| | L$_3$ of *S. papillosus* | 0.0 ± 0.0 [a] | 0.0 ± 0.0 [a] | 0.0 ± 0.0 [a] | 0.0 ± 0.0 [a] | 0.0 ± 0.0 [a] | – |
| | L$_3$ of *H. contortus* | 0.0 ± 0.0 [a] | 0.0 ± 0.0 [a] | 0.0 ± 0.0 [a] | 0.0 ± 0.0 [a] | 0.0 ± 0.0 [a] | – |
| 2-Pentanone | L$_{1–2}$ of *S. papillosus* | 0.0 ± 0.0 [a] | 33.3 ± 15.5 [b] | 0.0 ± 0.0 [a] | 0.0 ± 0.0 [a] | 0.0 ± 0.0 [a] | – |
| | L$_3$ of *S. papillosus* | 0.0 ± 0.0 [a] | 0.0 ± 0.0 [a] | 0.0 ± 0.0 [a] | 0.0 ± 0.0 [a] | 0.0 ± 0.0 [a] | – |
| | L$_3$ of *H. contortus* | 0.0 ± 0.0 [a] | 0.0 ± 0.0 [a] | 0.0 ± 0.0 [a] | 0.0 ± 0.0 [a] | 0.0 ± 0.0 [a] | – |

* LC—lethal concentration. [a, b, c, d]—different letters in the Table within each line indicate significant ($p < 0.05$) differences between groups according to the Tukey test results.

**Table 3.** Mortality of larvae of *S. papillosus* and *H. contortus* (%) during 24 h laboratory experiment under the influence of cyclic organic compounds (x ± SD, each experiment was repeated five times).

| Compound | Nematode Species | Mortality of Nematode Larvae in Control, % | Mortality of Nematode Larvae in 1.0% Solution, % | Mortality of Nematode Larvae in 0.1% Solution, % | Mortality of Nematode Larvae in 0.01% Solution, % | Mortality of Nematode Larvae in 0.001% Solution, % | LC$_{50}$, % * |
|---|---|---|---|---|---|---|---|
| Cyclopentanol | L$_{1–2}$ of *S. papillosus* | 0.0 ± 0.0 [a] | 16.9 ± 2.3 [b] | 6.0 ± 2.7 [c] | 0.0 ± 0.0 [a] | 0.0 ± 0.0 [a] | – |
| | L$_3$ of *S. papillosus* | 0.0 ± 0.0 [a] | 4.2 ± 5.8 [a] | 0.0 ± 0.0 [a] | 0.0 ± 0.0 [a] | 0.0 ± 0.0 [a] | – |
| | L$_3$ of *H. contortus* | 0.0 ± 0.0 [a] | 0.0 ± 0.0 [a] | 0.0 ± 0.0 [a] | 0.0 ± 0.0 [a] | 0.0 ± 0.0 [a] | – |
| Cyclohexanol | L$_{1–2}$ of *S. papillosus* | 0.0 ± 0.0 [a] | 100.0 ± 0.0 [b] | 48.1 ± 12.5 [c] | 0.0 ± 0.0 [a] | 0.0 ± 0.0 [a] | 0.1329 ± 0.2217 |
| | L$_3$ of *S. papillosus* | 0.0 ± 0.0 [a] | 100.0 ± 0.0 [b] | 37.7 ± 10.5 [c] | 0.0 ± 0.0 [a] | 0.0 ± 0.0 [a] | 0.2777 ± 0.1253 |
| | L$_3$ of *H. contortus* | 0.0 ± 0.0 [a] | 100.0 ± 0.0 [b] | 20.7 ± 13.1 [c] | 0.0 ± 0.0 [a] | 0.0 ± 0.0 [a] | 0.4325 ± 0.0964 |

**Table 3.** *Cont.*

| Compound | Nematode Species | Mortality of Nematode Larvae in Control, % | Mortality of Nematode Larvae in 1.0% Solution, % | Mortality of Nematode Larvae in 0.1% Solution, % | Mortality of Nematode Larvae in 0.01% Solution, % | Mortality of Nematode Larvae in 0.001% Solution, % | LC$_{50}$, % * |
|---|---|---|---|---|---|---|---|
| Cyclooctanone | L$_{1-2}$ of *S. papillosus* | 0.0 ± 0.0 [a] | 100.0 ± 0.0 [b] | 20.7 ± 3.3 [c] | 0.0 ± 0.0 [a] | 0.0 ± 0.0 [a] | 0.4325 ± 0.0237 |
| | L$_3$ of *S. papillosus* | 0.0 ± 0.0 [a] | 100.0 ± 0.0 [b] | 16.4 ± 4.1 [c] | 0.0 ± 0.0 [a] | 0.0 ± 0.0 [a] | 0.4617 ± 0.0265 |
| | L$_3$ of *H. contortus* | 0.0 ± 0.0 [a] | 47.3 ± 27.1 [b] | 0.0 ± 0.0 [a] | 0.0 ± 0.0 [a] | 0.0 ± 0.0 [a] | – |
| Phenol | L$_{1-2}$ of *S. papillosus* | 0.0 ± 0.0 [a] | 100.0 ± 0.0 [b] | 100.0 ± 0.0 [b] | 100.0 ± 0.0 [b] | 41.7 ± 8.9 [c] | 0.0023 ± 0.0012 |
| | L$_3$ of *S. papillosus* | 0.0 ± 0.0 [a] | 100.0 ± 0.0 [b] | 100.0 ± 0.0 [b] | 100.0 ± 0.0 [B] | 11.7 ± 16.2 [a] | 0.0049 ± 0.0010 |
| | L$_3$ of *H. contortus* | 0.0 ± 0.0 [a] | 100.0 ± 0.0 [b] | 100.0 ± 0.0 [b] | 6.7 ± 9.1 [a] | 0.0 ± 0.0 [a] | 0.0518 ± 0.0047 |
| Pyrocatechin | L$_{1-2}$ of *S. papillosus* | 0.0 ± 0.0 [a] | 100.0 ± 0.0 [b] | 100.0 ± 0.0 [b] | 65.8 ± 4.2 [c] | 9.4 ± 9.0 [a] | 0.0075 ± 0.0009 |
| | L$_3$ of *S. papillosus* | 0.0 ± 0.0 [a] | 100.0 ± 0.0 [b] | 100.0 ± 0.0 [b] | 10.2 ± 4.3 [c] | 0.0 ± 0.0 [a] | 0.0499 ± 0.0024 |
| | L$_3$ of *H. contortus* | 0.0 ± 0.0 [a] | 0.0 ± 0.0 [a] | 0.0 ± 0.0 [a] | 0.0 ± 0.0 [a] | 0.0 ± 0.0 [a] | – |
| Resorcinol | L$_{1-2}$ of *S. papillosus* | 0.0 ± 0.0 [a] | 100.0 ± 0.0 [b] | 100.0 ± 0.0 [b] | 82.8 ± 1.9 [c] | 6.9 ± 9.6 [a] | 0.0061 ± 0.0006 |
| | L$_3$ of *S. papillosus* | 0.0 ± 0.0 [a] | 100.0 ± 0.0 [b] | 24.5 ± 2.9 [c] | 6.9 ± 6.2 [d] | 0.0 ± 0.0 [a] | 0.4040 ± 0.0229 |
| | L$_3$ of *H. contortus* | 0.0 ± 0.0 [a] | 6.7 ± 9.1 [a] | 0.0 ± 0.0 [a] | 0.0 ± 0.0 [a] | 0.0 ± 0.0 [a] | – |
| Hydroquinone | L$_{1-2}$ of *S. papillosus* | 0.0 ± 0.0 [a] | 24.0 ± 8.2 [b] | 0.0 ± 0.0 [a] | 0.0 ± 0.0 [a] | 0.0 ± 0.0 [a] | – |
| | L$_3$ of *S. papillosus* | 0.0 ± 0.0 [a] | 0.0 ± 0.0 [a] | 0.0 ± 0.0 [a] | 0.0 ± 0.0 [a] | 0.0 ± 0.0 [a] | – |
| | L$_3$ of *H. contortus* | 0.0 ± 0.0 [a] | 0.0 ± 0.0 [a] | 0.0 ± 0.0 [a] | 0.0 ± 0.0 [a] | 0.0 ± 0.0 [a] | – |
| 2-naphthol | L$_{1-2}$ of *S. papillosus* | 0.0 ± 0.0 [a] | 100.0 ± 0.0 [b] | 100.0 ± 0.0 [b] | 100.0 ± 0.0 [b] | 27.7 ± 9.5 [c] | 0.0038 ± 0.0008 |
| | L$_3$ of *S. papillosus* | 0.0 ± 0.0 [a] | 100.0 ± 0.0 [b] | 100.0 ± 0.0 [b] | 100.0 ± 0.0 [b] | 3.3 ± 7.5 [a] | 0.0053 ± 0.0004 |
| | L$_3$ of *H. contortus* | 0.0 ± 0.0 [a] | 100.0 ± 0.0 [b] | 100.0 ± 0.0 [b] | 100.0 ± 0.0 [b] | 3.3 ± 7.5 [a] | 0.0053 ± 0.0004 |
| Anisole | L$_{1-2}$ of *S. papillosus* | 0.0 ± 0.0 [a] | 100.0 ± 0.0 [b] | 4.4 ± 4.8 [a] | 0.0 ± 0.0 [a] | 0.0 ± 0.0 [a] | 0.5293 ± 0.0237 |
| | L$_3$ of *S. papillosus* | 0.0 ± 0.0 [a] | 100.0 ± 0.0 [b] | 2.0 ± 4.5 [a] | 0.0 ± 0.0 [a] | 0.0 ± 0.0 [a] | 0.5408 ± 0.0211 |
| | L$_3$ of *H. contortus* | 0.0 ± 0.0 [a] | 96.7 ± 7.5 [b] | 0.0 ± 0.0 [a] | 0.0 ± 0.0 [a] | 0.0 ± 0.0 [a] | 0.5654 ± 0.0363 |
| Phenyl ether | L$_{1-2}$ of *S. papillosus* | 0.0 ± 0.0 [a] | 100.0 ± 0.0 [b] | 100.0 ± 0.0 [b] | 100.0 ± 0.0 [b] | 47.2 ± 9.9 [c] | 0.0015 ± 0.0017 |
| | L$_3$ of *S. papillosus* | 0.0 ± 0.0 [a] | 100.0 ± 0.0 [b] | 100.0 ± 0.0 [b] | 77.3 ± 6.6 [c] | 0.0 ± 0.0 [a] | 0.0068 ± 0.0005 |
| | L$_3$ of *H. contortus* | 0.0 ± 0.0 [a] | 70.0 ± 24.0 [b] | 11.7 ± 16.2 [a] | 9.0 ± 12.4 [a] | 0.0 ± 0.0 [a] | 0.6913 ± 0.3352 |
| Ortho-dimethylbenzene | L$_{1-2}$ of *S. papillosus* | 0.0 ± 0.0 [a] | 28.0 ± 5.5 [b] | 0.0 ± 0.0 [a] | 0.0 ± 0.0 [a] | 0.0 ± 0.0 [a] | – |
| | L$_3$ of *S. papillosus* | 0.0 ± 0.0 [a] | 13.0 ± 7.7 [b] | 0.0 ± 0.0 [a] | 0.0 ± 0.0 [a] | 0.0 ± 0.0 [a] | – |
| | L$_3$ of *H. contortus* | 0.0 ± 0.0 [a] | 0.0 ± 0.0 [a] | 0.0 ± 0.0 [a] | 0.0 ± 0.0 [a] | 0.0 ± 0.0 [a] | – |
| Piperonyl alcohol | L$_{1-2}$ of *S. papillosus* | 0.0 ± 0.0 [a] | 100.0 ± 0.0 [b] | 82.5 ± 7.5 [c] | 37.6 ± 1.7 [d] | 0.0 ± 0.0 [a] | 0.0349 ± 0.0067 |
| | L$_3$ of *S. papillosus* | 0.0 ± 0.0 [a] | 100.0 ± 0.0 [b] | 6.4 ± 5.9 [c] | 0.0 ± 0.0 [a] | 0.0 ± 0.0 [a] | 0.5192 ± 0.0304 |
| | L$^3$ of *H. contortus* | 0.0 ± 0.0 [a] | 28.3 ± 18.3 [b] | 0.0 ± 0.0 [a] | 3.3 ± 7.5 [a] | 0.0 ± 0.0 [a] | – |
| 4-Hydroxy-3-methoxy benzyl alcohol | L$_{1-2}$ of *S. papillosus* | 0.0 ± 0.0 [a] | 100.0 ± 0.0 [b] | 100.0 ± 0.0 [b] | 100.0 ± 0.0 [b] | 17.0 ± 9.7 [c] | 0.0046 ± 0.0006 |
| | L$_3$ of *S. papillosus* | 0.0 ± 0.0 [a] | 36.0 ± 6.3 [b] | 14.2 ± 6.1 [c] | 0.0 ± 0.0 [a] | 0.0 ± 0.0 [a] | – |
| | L$_3$ of *H. contortus* | 2.9 ± 6.4 [a] | 6.7 ± 9.1 [a] | 3.3 ± 7.5 [a] | 0.0 ± 0.0 [a] | 0.0 ± 0.0 [a] | – |

**Table 3.** *Cont.*

| Compound | Nematode Species | Mortality of Nematode Larvae in Control, % | Mortality of Nematode Larvae in 1.0% Solution, % | Mortality of Nematode Larvae in 0.1% Solution, % | Mortality of Nematode Larvae in 0.01% Solution, % | Mortality of Nematode Larvae in 0.001% Solution, % | LC$_{50}$, % * |
|---|---|---|---|---|---|---|---|
| Gallic acid-1-hydrate | L$_{1-2}$ of *S. papillosus* | 0.0 ± 0.0 [a] | 18.2 ± 4.9 [b] | 0.0 ± 0.0 [a] | 0.0 ± 0.0 [a] | 0.0 ± 0.0 [a] | – |
| | L$_3$ of *S. papillosus* | 0.0 ± 0.0 [a] | 0.0 ± 0.0 [a] | 0.0 ± 0.0 [a] | 0.0 ± 0.0 [a] | 0.0 ± 0.0 [a] | – |
| | L$_3$ of *H. contortus* | 0.0 ± 0.0 [a] | 0.0 ± 0.0 [a] | 0.0 ± 0.0 [a] | 0.0 ± 0.0 [a] | 0.0 ± 0.0 [a] | – |
| Dibutyl phthalate | L$_{1-2}$ of *S. papillosus* | 0.0 ± 0.0 [a] | 22.0 ± 10.0 [b] | 0.0 ± 0.0 [a] | 0.0 ± 0.0 [a] | 0.0 ± 0.0 [a] | – |
| | L$_3$ of *S. papillosus* | 0.0 ± 0.0 [a] | 0.0 ± 0.0 [a] | 0.0 ± 0.0 [a] | 0.0 ± 0.0 [a] | 0.0 ± 0.0 [a] | – |
| | L$_3$ of *H. contortus* | 0.0 ± 0.0 [a] | 0.0 ± 0.0 [a] | 0.0 ± 0.0 [a] | 0.0 ± 0.0 [a] | 0.0 ± 0.0 [a] | – |
| 3-Furoic acid | L$_{1-2}$ of *S. papillosus* | 0.0 ± 0.0 [a] | 100.0 ± 0.0 [b] | 100.0 ± 0.0 [b] | 11.3 ± 8.2 [c] | 0.0 ± 0.0 [a] | 0.0493 ± 0.0047 |
| | L$_3$ of *S. papillosus* | 0.0 ± 0.0 [a] | 100.0 ± 0.0 [b] | 100.0 ± 0.0 [b] | 19.5 ± 6.2 [c] | 0.0 ± 0.0 [a] | 0.0441 ± 0.0043 |
| | L$_3$ of *H. contortus* | 0.0 ± 0.0 [a] | 96.7 ± 7.5 [b] | 0.0 ± 0.0 [a] | 0.0 ± 0.0 [a] | 0.0 ± 0.0 [a] | 0.5654 ± 0.0363 |
| Succinic anhydride | L$_{1-2}$ of *S. papillosus* | 0.0 ± 0.0 [a] | 48.5 ± 21.5 [b] | 0.0 ± 0.0 [a] | 0.0 ± 0.0 [a] | 0.0 ± 0.0 [a] | – |
| | L$_3$ of *S. papillosus* | 0.0 ± 0.0 [a] | 0.0 ± 0.0 [a] | 0.0 ± 0.0 [a] | 0.0 ± 0.0 [a] | 0.0 ± 0.0 [a] | – |
| | L$_3$ of *H. contortus* | 0.0 ± 0.0 [a] | 0.0 ± 0.0 [a] | 0.0 ± 0.0 [a] | 0.0 ± 0.0 [a] | 0.0 ± 0.0 [a] | – |
| Maleic anhydrid | L$_{1-2}$ of *S. papillosus* | 0.0 ± 0.0 [a] | 100.0 ± 0.0 [b] | 100.0 ± 0.0 [b] | 28.2 ± 2.6 [c] | 0.0 ± 0.0 [a] | 0.0373 ± 0.0023 |
| | L$_3$ of *S. papillosus* | 0.0 ± 0.0 [a] | 100.0 ± 0.0 [b] | 100.0 ± 0.0 [b] | 13.3 ± 18.3 [c] | 0.0 ± 0.0 [a] | 0.0481 ± 0.0115 |
| | L$_3$ of *H. contortus* | 0.0 ± 0.0 [a] | 63.2 ± 26.9 [b] | 0.0 ± 0.0 [a] | 0.0 ± 0.0 [a] | 0.0 ± 0.0 [a] | 0.8120 ± 0.3701 |
| 5-Methylfurfural | L$_{1-2}$ of *S. papillosus* | 0.0 ± 0.0 [a] | 100.0 ± 0.0 [b] | 87.2 ± 2.8 [c] | 51.8 ± 5.8 [d] | 13.6 ± 5.3 [e] | 0.0096 ± 0.0014 |
| | L$_3$ of *S. papillosus* | 0.0 ± 0.0 [a] | 100.0 ± 0.0 [b] | 71.1 ± 11.5 [c] | 25.2 ± 10.0 [d] | 0.0 ± 0.0 [a] | 0.0586 ± 0.0212 |
| | L$_3$ of *H. contortus* | 0.0 ± 0.0 [a] | 100.0 ± 0.0 [b] | 10.0 ± 5.0 [c] | 0.0 ± 0.0 [a] | 0.0 ± 0.0 [a] | 0.5000 ± 0.0279 |
| 2-Methylfuran | L$_{1-2}$ of *S. papillosus* | 0.0 ± 0.0 [a] | 31.2 ± 5.9 [b] | 0.0 ± 0.0 [a] | 0.0 ± 0.0 [a] | 0.0 ± 0.0 [a] | – |
| | L$_3$ of *S. papillosus* | 0.0 ± 0.0 [a] | 16.1 ± 11.3 [b] | 0.0 ± 0.0 [a] | 0.0 ± 0.0 [a] | 0.0 ± 0.0 [a] | – |
| | L$_3$ of *H. contortus* | 0.0 ± 0.0 [a] | 8.3 ± 11.8 [a] | 0.0 ± 0.0 [a] | 0.0 ± 0.0 [a] | 0.0 ± 0.0 [a] | – |
| Furfuryl alcohol | L$_{1-2}$ of *S. papillosus* | 0.0 ± 0.0 [a] | 100.0 ± 0.0 [b] | 82.5 ± 4.9 [c] | 12.9 ± 3.2 [d] | 0.0 ± 0.0 [a] | 0.0580 ± 0.0053 |
| | L$_3$ of *S. papillosus* | 0.0 ± 0.0 [a] | 100.0 ± 0.0 [b] | 2.9 ± 6.4 [a] | 2.5 ± 5.6 [a] | 0.0 ± 0.0 [a] | 0.5366 ± 0.0307 |
| | L$_3$ of *H. contortus* | 0.0 ± 0.0 [a] | 0.0 ± 0.0 [a] | 0.0 ± 0.0 [a] | 0.0 ± 0.0 [a] | 0.0 ± 0.0 [a] | – |

* LC—lethal concentration. [a, b, c, d]—different letters in the Table within each line indicate significant ($p < 0.05$) differences between groups according to the Tukey test results.

**Table 4.** Mortality of larvae of *S. papillosus*, *H. contortus* (%) during 24 h laboratory experiment under the influence of sulfur- and nitrogen-containing organic compounds (x ± SD, each experiment was repeated five times).

| Compounds | Nematode Species | Mortality of Nematode Larvae in Control, % | Mortality of Nematode Larvae in 1.0% Solution, % | Mortality of Nematode Larvae in 0.1% Solution, % | Mortality of Nematode Larvae in 0.01% Solution, % | Mortality of Nematode Larvae in 0.001% Solution, % | LC$_{50}$, % * |
|---|---|---|---|---|---|---|---|
| Thioacetic acid | L$_{1-2}$ of *S. papillosus* | 0.0 ± 0.0 [a] | 100.0 ± 0.0 [b] | 100.0 ± 0.0 [b] | 100.0 ± 0.0 [b] | 45.8 ± 16.9 [c] | 0.0017 ± 0.0029 |
| | L$_3$ of *S. papillosus* | 0.0 ± 0.0 [a] | 100.0 ± 0.0 [b] | 100.0 ± 0.0 [b] | 98.2 ± 2.4 [b] | 11.7 ± 16.2 [c] | 0.0050 ± 0.0011 |
| | L$_3$ of *H. contortus* | 0.0 ± 0.0 [a] | 100.0 ± 0.0 [b] | 100.0 ± 0.0 [b] | 41.7 ± 11.8 [c] | 0.0 ± 0.0 [a] | 0.0228 ± 0.0163 |

**Table 4.** *Cont.*

| Compounds | Nematode Species | Mortality of Nematode Larvae in Control, % | Mortality of Nematode Larvae in 1.0% Solution, % | Mortality of Nematode Larvae in 0.1% Solution, % | Mortality of Nematode Larvae in 0.01% Solution, % | Mortality of Nematode Larvae in 0.001% Solution, % | LC$_{50}$, % * |
|---|---|---|---|---|---|---|---|
| Taurine | L$_{1-2}$ of *S. papillosus* | 0.0 ± 0.0 [a] | 5.0 ± 8.0 [a] | 0.0 ± 0.0 [a] | 0.0 ± 0.0 [a] | 0.0 ± 0.0 [a] | – |
| | L$_3$ of *S. papillosus* | 0.0 ± 0.0 [a] | 0.0 ± 0.0 [a] | 0.0 ± 0.0 [a] | 0.0 ± 0.0 [a] | 0.0 ± 0.0 [a] | – |
| | L$_3$ of *H. contortus* | 0.0 ± 0.0 [a] | 0.0 ± 0.0 [a] | 0.0 ± 0.0 [a] | 0.0 ± 0.0 [a] | 0.0 ± 0.0 [a] | – |
| Butan-1-amine | L$_{1-2}$ of *S. papillosus* | 0.0 ± 0.0 [a] | 100.0 ± 0.0 [b] | 82.2 ± 3.5 [ca] | 0.0 ± 0.0 [a] | 0.0 ± 0.0 [a] | 0.0647 ± 0.0023 |
| | L$_3$ of *S. papillosus* | 0.0 ± 0.0 [a] | 100.0 ± 0.0 [b] | 10.6 ± 2.5 [c] | 0.0 ± 0.0 [a] | 0.0 ± 0.0 [a] | 0.4966 ± 0.0141 |
| | L$_3$ of *H. contortus* | 0.0 ± 0.0 [a] | 100.0 ± 0.0 [b] | 5.0 ± 8.5 [a] | 0.0 ± 0.0 [a] | 0.0 ± 0.0 [a] | 0.5263 ± 0.0427 |
| 6-Aminocaproic acid | L$_{1-2}$ of *S. papillosus* | 0.0 ± 0.0 [a] | 0.0 ± 0.0 [a] | 0.0 ± 0.0 [a] | 0.0 ± 0.0 [a] | 0.0 ± 0.0 [a] | – |
| | L$_3$ of *S. papillosus* | 0.0 ± 0.0 [a] | 0.0 ± 0.0 [a] | 0.0 ± 0.0 [a] | 0.0 ± 0.0 [a] | 0.0 ± 0.0 [a] | – |
| | L$_3$ of *H. contortus* | 0.0 ± 0.0 [a] | 0.0 ± 0.0 [a] | 0.0 ± 0.0 [a] | 0.0 ± 0.0 [a] | 0.0 ± 0.0 [a] | – |
| Dimethylformamide | L$_{1-2}$ of *S. papillosus* | 0.0 ± 0.0 [a] | 72.1 ± 6.6 [b] | 9.6 ± 2.9 [c] | 0.0 ± 0.0 [a] | 0.0 ± 0.0 [a] | 0.6818 ± 0.0765 |
| | L$_3$ of *S. papillosus* | 0.0 ± 0.0 [a] | 12.7 ± 3.8 [b] | 0.0 ± 0.0 [a] | 0.0 ± 0.0 [a] | 0.0 ± 0.0 [a] | – |
| | L$^3$ of *H. contortus* | 0.0 ± 0.0 [a] | 0.0 ± 0.0 [a] | 0.0 ± 0.0 [a] | 0.0 ± 0.0 [a] | 0.0 ± 0.0 [a] | – |
| Glutamic acid | L$_{1-2}$ of *S. papillosus* | 0.0 ± 0.0 [a] | 5.0 ± 5.0 [a] | 0.0 ± 0.0 [a] | 0.0 ± 0.0 [a] | 0.0 ± 0.0 [a] | – |
| | L$_3$ of *S. papillosus* | 0.0 ± 0.0 [a] | 0.0 ± 0.0 [a] | 0.0 ± 0.0 [a] | 0.0 ± 0.0 [a] | 0.0 ± 0.0 [a] | – |
| | L$_3$ of *H. contortus* | 0.0 ± 0.0 [a] | 0.0 ± 0.0 [a] | 0.0 ± 0.0 [a] | 0.0 ± 0.0 [a] | 0.0 ± 0.0 [a] | – |
| Carnitine | L$_{1-2}$ of *S. papillosus* | 0.0 ± 0.0 [a] | 0.0 ± 0.0 [a] | 0.0 ± 0.0 [a] | 0.0 ± 0.0 [a] | 0.0 ± 0.0 [a] | – |
| | L$_3$ of *S. papillosus* | 0.0 ± 0.0 [a] | 0.0 ± 0.0 [a] | 0.0 ± 0.0 [a] | 0.0 ± 0.0 [a] | 0.0 ± 0.0 [a] | – |
| | L$_3$ of *H. contortus* | 0.0 ± 0.0 [a] | 0.0 ± 0.0 [a] | 0.0 ± 0.0 [a] | 0.0 ± 0.0 [a] | 0.0 ± 0.0 [a] | – |
| Ornithine monohydrochloride | L$_{1-2}$ of *S. papillosus* | 0.0 ± 0.0 [a] | 10.0 ± 12.5 [a] | 0.0 ± 0.0 [a] | 0.0 ± 0.0 [a] | 0.0 ± 0.0 [a] | – |
| | L$_3$ of *S. papillosus* | 0.0 ± 0.0 [a] | 0.0 ± 0.0 [a] | 0.0 ± 0.0 [a] | 0.0 ± 0.0 [a] | 0.0 ± 0.0 [a] | – |
| | L$_3$ of *H. contortus* | 0.0 ± 0.0 [a] | 0.0 ± 0.0 [a] | 0.0 ± 0.0 [a] | 0.0 ± 0.0 [a] | 0.0 ± 0.0 [a] | – |
| 1-Phenylethan-1-amine | L$_{1-2}$ of *S. papillosus* | 0.0 ± 0.0 [a] | 100.0 ± 0.0 [b] | 100.0 ± 0.0 [b] | 29.8 ± 2.2 [c] | 0.0 ± 0.0 [a] | 0.0359 ± 0.0020 |
| | L$_3$ of *S. papillosus* | 0.0 ± 0.0 [a] | 100.0 ± 0.0 [b] | 100.0 ± 0.0 [b] | 29.1 ± 5.0 [c] | 0.0 ± 0.0 [a] | 0.0365 ± 0.0045 |
| | L$_3$ of *H. contortus* | 0.0 ± 0.0 [a] | 100.0 ± 0.0 [b] | 21.3 ± 19.7 [c] | 0.0 ± 0.0 [a] | 0.0 ± 0.0 [a] | 0.4282 ± 0.1527 |
| 3-Aminobenzoic acid | L$_{1-2}$ of *S. papillosus* | 0.0 ± 0.0 [a] | 100.0 ± 0.0 [b] | 15.9 ± 5.6 [c] | 0.0 ± 0.0 [a] | 0.0 ± 0.0 [a] | 0.4649 ± 0.0358 |
| | L$_3$ of *S. papillosus* | 0.0 ± 0.0 [a] | 87.1 ± 7.3 [b] | 6.2 ± 8.5 [a] | 0.0 ± 0.0 [a] | 0.0 ± 0.0 [a] | 0.5873 ± 0.0874 |
| | L$_3$ of *H. contortus* | 0.0 ± 0.0 [a] | 5.0 ± 11.2 [a] | 0.0 ± 0.0 [a] | 0.0 ± 0.0 [a] | 0.0 ± 0.0 [a] | – |

**Table 4.** *Cont.*

| Compounds | Nematode Species | Mortality of Nematode Larvae in Control, % | Mortality of Nematode Larvae in 1.0% Solution, % | Mortality of Nematode Larvae in 0.1% Solution, % | Mortality of Nematode Larvae in 0.01% Solution, % | Mortality of Nematode Larvae in 0.001% Solution, % | $LC_{50}$, % * |
|---|---|---|---|---|---|---|---|
| 2-Methyl-5-nitroimidazole | $L_{1-2}$ of *S. papillosus* | $0.0 \pm 0.0$ [a] | $0.0 \pm 0.0$ [a] | $0.0 \pm 0.0$ [a] | $0.0 \pm 0.0$ [a] | $0.0 \pm 0.0$ [a] | – |
| | $L_3$ of *S. papillosus* | $0.0 \pm 0.0$ [a] | $0.0 \pm 0.0$ [a] | $0.0 \pm 0.0$ [a] | $0.0 \pm 0.0$ [a] | $0.0 \pm 0.0$ [a] | – |
| | $L_3$ of *H. contortus* | $0.0 \pm 0.0$ [a] | $0.0 \pm 0.0$ [a] | $0.0 \pm 0.0$ [a] | $0.0 \pm 0.0$ [a] | $0.0 \pm 0.0$ [a] | – |

* LC—lethal concentration. [a, b, c]—different letters in the Table within each line indicate significant ($p < 0.05$) differences between groups according to the Tukey test results.

Over 90% of all the examined species of larvae of various development stages died in the in vitro experiments under the effects of ethyl pyruvate (Table 2). Tert butyl carboxylic acid, 3,7-dimethyl-6-octanoic acid, isobutanaldehyde, phenyl ether, butyl acrylate, maleic anhydride, 1-phenylethan-1-amine appeared to be less toxic to *H. contortus* of the third (invasive) stage. Over 24 h, in 1% solution, over 60% of the larvae of this species died (Tables 2–4).

Of the acyclic organic compounds, the lowest $LC_{50}$ parameters for the non-invasive larvae and invasive larvae of *S. papillosus* were produced by 2-oxopentanedioic acid, *S. papillosus*, and by diethyl malonate and 2-oxopentanedioic acid for the *H. contortus* larvae (Table 2).

Similar results were produced by the influence of phenol (carbolic acid) on the non-invasive and invasive nematode larvae: $LC_{50}$ of this compound did not exceed 0.0049 for *S. papillosus* and 0.0518 for *H. contortus* (Table 3). However, stronger effects on the nematode larvae of various stages of development were exerted by cyclic organic compound 2-naphthol (Table 3).

Of the sulfur- and nitrogen-containing organic compounds, notable negative impacts were displayed by thioacetic acid and hexylamine (Table 4). Over 90% of the *S. papillosus* larvae died even in 0.01% thioacetic acid solution. The most thioacetic acid-resistant larvae were observed to be *H. contortus*. We saw 100% death of the larvae of this nematode species in the exposure to 0.1% concentration of thioacetic acid.

The weakest effects on the nematode larvae of various development stages were exerted by 6-aminocaproic acid, butyl glycol, gallic acid-1-hydrate, hydroquinone, glutamic acid, methyl-2-nitroimidazole-5, dibutyl phthalate, carnitine, octadecanol-1, oleic acid, ornithine monohydrochloride, propylene glycol-1,2, stearyl alcohol, taurine, succinic anhydride, 4-methyl-2-pentanol, maleic acid, 2-pentanone, methanol, phytol, propan-2-ol. All the invasive *S. papillosus* and *H. contortus* larvae survived exposure to 1% solutions of those compounds. Additionally, over 70% of the non-invasive development stages of *S. papillosus* remained vital for 24 h after being subject to the same concentration of the organic compounds (Table 2).

## 4. Discussion

The obtained results indicate notable nematocidal properties of 28 compounds: 1-phenylethan-1-amine, 2-methylbutanoic acid, 2-oxopentanedioic acid, 3,7-dimethyl-6-octenoic acid, 3-furoic acid, 5-methylfurfural, allyl acetoacetate, anisole, butan-1-amine, butyl acrylate, cyclohexanol, diethyl malonate, ethyl acetoacetate, ethyl pyruvate, glutaraldehyde, isobutyraldehyde, isovaleric acid, maleic anhydrid, methyl acetoacetate, naphthol-2, phenol, phenyl ether, piperonyl alcohol, pyrocatechin, resorcinol, tert butyl carboxylic acid, hexylamine, and thioacetic acid.

Glutaraldehyde is an organic compound, the properties of which are being researched all around the globe [33]. The efficiency of glutaraldehyde was confirmed against fungi, bacteria, and viruses [34,35]. According to the results of our studies, a 1% concentration of

this compound was toxic to the larvae of *S. papillosus* and *H. contortus*. Further research on its anthelmintic properties is of great interest to veterinary specialists and agronomists for the purposes of designing treatment and prophylaxis measures in livestock enterprises and for combating nematodes that are pests to agricultural plants.

Chitwood [36] described the antagonistic activities of compounds present in plants, including phenols, against nematodes that are pests of agricultural plants. The results of our studies of nematocidal properties of phenol against nematode larvae that are pests of agricultural animals indicate the negative effect of this compound as well. Its 0.1% solution killed the *S. papillosus* and *H. contortus* larvae of all stages in 24 h.

The great potential of plant compounds in combating plant nematodes was also described by Andrés et al. [37]. They determined that these compounds may be used as nematocides and be included as components of complex pesticide mixtures with increased efficacy. Ajith et al. [38] studied the nematocidal properties of eugenol—one of the main components of *Ocimum* and *Dianthus* essential oils—against *Meloidogyne graminicola* (Golden and Birchfield, 1965). Our previous studies of the nematocidal potentials of essential oils indicate that the essential oil *Syzygium aromaticum* (L.) has a toxic effect on nematode larvae of ruminants in in vitro conditions [39].

Stavropoulou et al. [40] also report the toxic activity of eugenol towards bulb nematodes *Ditylenchus dipsaci* (Kühn, 1857) isolated from infested garlic cloves. Helal et al. [41] report anthelmintic properties of coriander extract (which contains eugenol) on third-stage larvae of ruminant nematodes *H. contortus*, *Trichostrongylus axei* (Cobbold, 1879), *T. colubriformis* (Giles, 1892), *T. vitrines* (Nisbet and Gasser, 2004), *Teladorsagia circumcincta* (Stadelman, 1894) and *Cooperia oncophora* (Railliet, 1898). Silva et al. [42] expressed great concern regarding the tolerance of *H. contortus* to synthetic anthelmintic drugs. These scientists determined that the greatest effect exerted by plant monoterpenes against this nematode species was exhibited by carvacrol ($IC_{50}$ = 185.9 µg/mL) and thymol ($IC_{50}$ = 187.0 µg/mL).

Essential oils have been found to be a new source of human- and environment-safe compounds that have nematocidal activity towards pests of agricultural crops, including nematodes that are plant parasites, as reported by Avato et al. [43], Eloh et al. [44], D'Addabbo and Avato [45], and Douda et al. [46]. Earlier, we reported the nematocidal properties of some organic acids and also other organic compounds [47].

Therefore, many organic compounds present in cells of living organisms (plants, fungi, animals, including nematodes) cause no negative impact (phytol, 3-hydroxy-2-butanone, maleic acid, oleic acid, hydroquinone, gallic acid-1-hydrate, taurine, 6-aminocaproic acid, glutamic acid, carnitine, ornithine monohydrochloride) on the vitality of parasitic nematodes even in 10 g/L concentrations (i.e., in 1% solution, tested in our experiment). Despite being abundant in nature, some of those compounds can be lethal to nematodes (3,7-dimethyl-6-octenoic acid, isovaleric acid, glycolic acid, 2-oxopentanedioic acid, 2-methylbutanoic acid, anisole, 4-hydroxy-3-methoxybenzyl alcohol, furfuryl alcohol). Those particular compounds are of the greatest interest for ecologically clean control of nematodes in the conditions of maintenance of animals on-premises and on farms.

Some of the compounds we tested are chemically synthesized xenobiotics. They are introduced into the soil with industrial and municipal wastes. According to the results of our studies, these compounds could pose a serious threat to nematode larvae (glutaraldehyde, 1,4-diethyl 2-methyl-3-oxobutanedioate, hexylamine, diethyl malonate, allyl acetoacetate, tert butyl carboxylic acid, butyl acrylate, 3-methyl-2-butanone, isobutyraldehyde, methyl acetoacetate, ethyl acetoacetate, ethyl pyruvate, 3-methylbutanal, cyclohexanol, cyclooctanone, phenol, pyrocatechin, resorcinol, naphthol-2, phenyl ether, piperonyl alcohol, 3-furoic acid, maleic anhydrid, 5-methylfurfural, thioacetic acid, butan-1-amine, dimethylformamide, 1-phenylethan-1-amine, 3-aminobenzoic acid). Apparently, some of these xenobiotics had no negative impact on the larvae of the parasitic nematodes (methanol, propan-2-ol, isoamyl alcohol, propylene glycol-1,2, octadecanol-1, 4-methyl-2-pentanol, 2-ethoxyethanol, butyl glycol, 2-pentanone, cyclopentanol, ortho-dimethylbenzene, dibutyl phthalate, succinic anhydride, 2-methylfuran, 2-methyl-5-nitroimidazole).

The lowest $LC_{50}$ of all the compounds we tested were observed for 28 of them: 1-phenylethan-1-amine, 2-methylbutanoic acid, 2-oxopentanedioic acid, 3,7-dimethyl-6-octenoic acid, 3-furoic acid, 5-methylfurfural, allyl acetoacetate, anisole, butan-1-amine, butyl acrylate, cyclohexanol, diethyl malonate, ethyl acetoacetate, ethyl pyruvate, glutaraldehyde, isobutyraldehyde, isovaleric acid, maleic anhydrid, methyl acetoacetate, naphthol-2, phenol, phenyl ether, piperonyl alcohol, pyrocatechin, resorcinol, tert butyl carboxylic acid, hexylamine, thioacetic acid. Since most of them are toxic to vertebrates and invertebrates (Table 5), they could not be applied in the natural conditions of pastures. However, 3,7-dimethyl-6-octenoic acid, 5-methylfurfural, anisole, cyclohexanol, diethyl malonate, ethyl acetoacetate, methyl acetoacetate, and phenyl ether could be used for further experiments in livestock premises, because they exhibit sufficiently low-toxicity to vertebrates and are efficient in killing nematodes (Table 5).

**Table 5.** Nematocidal activity for larvae of *S. papillosus* and *H. contortus* (our data) and toxicity * to vertebrate animals (rats and mice *, the literature data, https://pubchem.ncbi.nlm.nih.gov accessed on 10 January 2023).

| Compounds | $LD_{50}$ for Rats ***, mg/kg (oral) | $LD_{50}$ for Mice ***, mg/kg (oral) | $LC_{50}$ for $L_{1-2}$ of *S. papillosus* ****, mg/kg | $LC_{50}$ for $L_3$ of *S. papillosus* ****, mg/kg | $LC_{50}$ for $L_3$ of *H. contortus* ****, mg/kg |
|---|---|---|---|---|---|
| 1-Phenylethan-1-amine | 940 | 560 | 359 | 365 | 4282 |
| 2-Methylbutanoic acid | 4100 | 4550 | 20 | 52 | >10,000 |
| 2-Oxopentanedioic acid | – | – | 19 | 39 | 289 |
| 3,7-Dimethyl-6-octenoic acid | 2610 | – | 520 | 825 | 6140 |
| 3-Furoic acid | – | – | 493 | 441 | 5654 |
| 5-Methylfurfural | 2200 | – | 96 | 586 | 5000 |
| Allyl acetoacetate | – | – | 73 | 2525 | 6799 |
| Anisole | 3700 | 2800 | 5293 | 5408 | 5654 |
| Butan-1-amine | 366 | – | 647 | 4966 | 5263 |
| Butyl acrylate | 900 | – | 121 | 360 | 4757 |
| Cyclohexanol | 1400 | – | 1329 | 2777 | 4325 |
| Diethyl malonate | 14,900 | 6400 | 435 | 452 | 146 |
| Ethyl acetoacetate | 3980 | 5105 | 64 | 3353 | 5055 |
| Ethyl pyruvate | – | – | 1935 | 5823 | 5854 |
| Glutaraldehyde | 134 | 100 | 784 | 1346 | 5166 |
| Hexylamine | 670 | – | 56 | 383 | 4375 |
| Isobutyraldehyde | 960 | – | 527 | 696 | 6726 |
| Isovaleric acid | 2000 | – | 487 | 626 | 5408 |
| Maleic anhydrid | 400 | 465 | 373 | 481 | 8120 |
| Methyl acetoacetate | 3228 | – | 592 | 4598 | >10,000 |
| Naphthol-2 | 1870 | 275 | 38 | 53 | 53 |
| Phenol | 317 | 270 | 23 | 49 | 518 |
| Phenyl ether | 2450 | – | 15 | 68 | 6913 |
| Piperonyl alcohol | – | – | 349 | 5192 | >10,000 |
| Pyrocatechin | 260 | 260 | 75 | 499 | >10,000 |
| Resorcinol | 301 | 200 | 61 | 4040 | >10,000 |
| Tert butyl carboxylic acid | – | – | 681 | 5243 | 6402 |
| Thioacetic acid | – | – | 17 | 50 | 228 |

* *Rattus* Fischer, 1803, *Mus* Linnaeus, 1766, **—dash indicates absence of scientific data regarding $LD_{50}$ for rats or mice. *** LD—lethal dose, **** LC—lethal concentration.

## 5. Conclusions

Thus, the compounds that in 1% solutions displayed lethal actions towards the free-living stages of the nematode larvae of *S. papillosus* and *H. contortus*, are promising for further experiments.

Organic compounds that are used in various spheres of human activity and often occur in nature can exhibit appreciable nematocidal properties. The results we obtained could be used for combating invasive nematode larvae that are parasites of agricultural animals in the environment in order to decrease toxic pesticide loading on natural ecosystems.

**Author Contributions:** Conceptualization, O.B. and V.B.; methodology, O.B.; validation, V.B.; formal analysis, V.B.; investigation, O.B.; resources, O.B. and V.B.; data curation, O.B. and V.B.; writing—original draft preparation, O.B. and V.B.; writing—review and editing, O.B. and V.B.; visualization, O.B. and V.B.; supervision, O.B. and V.B.; project administration, O.B.; funding acquisition, O.B. All authors have read and agreed to the published version of the manuscript.

**Funding:** This research was funded by the Ministry of Education and Science of Ukraine, grant number 0120U102384 "Evaluation of antiparasitic properties of medicinal plants in livestock production" (2023–2025).

**Institutional Review Board Statement:** Not applicable.

**Data Availability Statement:** The data presented in this study are available on request from the corresponding author.

**Conflicts of Interest:** The authors declare no conflict of interest.

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
