# Peer review of "Survival of Nematode Larvae Strongyloides papillosus and Haemonchus contortus under the Influence of Various Groups of Organic Compounds"

_diversity, doi:10.3390/d15020254_

Round 1

Reviewer 1 Report

The manuscript is about checking the survival of two nematode species under the influence of various compounds. The data obtained are interesting, but the paper needs some major revisions.

 Key words:

in order not to repeat the words from the title maybe instead of nematode species will be better to specify the families of these nematodes

Introduction:

„Nematodes have no eyesight or hearing” – this is obvious, but it looks/sounds very strange - I propose to delete this sentence.

Material and methods:

There is no information on how many nematodes (by species and life stages) were used for this experiments. Only the volume of sediment is given, but here the number of nematodes is important; after the experiment the nematodes were counted so the authors surely have this data. 

Whole species/nematode names i.e. with authors and date (Strongyloides papillosus, Haemonchus contortus) are already given in the introduction, so repeating them in subsequent chapters is unnecessary. When first time (abstract and main text) a species is mentioned in the text, its whole name should be given, that is, the common (English) name, the scientific name with the author(s) and the date.

By the way, please complete the other species names (author/authors, date) used in the text, e.g. Meloidogyne javanica, etc, etc.

A 0.001% solution of was also used ?, see Tables 2-5.

Tables 2-5:

The notation "n=5" is inaccurate; this is, after all, the number of repeatitions, not the number of nematodes.

Table 5:

The authors mention "rats" and "mice" - but specifically which species are involved. The common names are very vague - it is unclear what species were used in the experiment; it needs to be clarified/specified whether it is the burnished enggano rat Rattus adustus Sody, 1940, or perhaps the Australian swamp rat Rattus lutreolus (J. E. Gray, 1841) or the brown rat/ Norway rat Rattus norvegicus (Berkenhout, 1769), or the house rat/black rat Rattus rattus (Linnaeus, 1758).

Using only common names is very confusing, especially since some species (e.g., rats) have several common names each, such as the aforementioned R. norvegicus and R. rattus.

Unfortunately, it can happen that in the cited articles/databases this (species names) is not specified, and this is a serious mistake; often in veterinary works such errors are very common.

So, in addition to the common names, you need to add scientific names (genera or species) to the title of this table.

The comment also applies to the names listed in the Introduction section - mice, rats, cats, ets.

Please explain the abbreviations LC and LD in the legend

Discusion and colnculison:

These enumerations of compounds are very tiresome; they are repetitions; I doubt whether there is any point in listing them in the Discussion; in any case, they certainly need to be deleted from the Conclusion chapter.

Author Response

Dear Reviewer,

Thank you very much for the work done with our article. We have tried to correct all your comments.

Key words:

in order not to repeat the words from the title maybe instead of nematode species will be better to specify the families of these nematodes

Done

Introduction: „Nematodes have no eyesight or hearing” – this is obvious, but it looks/sounds very strange - I propose to delete this sentence.

Done

Material and methods: There is no information on how many nematodes (by species and life stages) were used for this experiments. Only the volume of sediment is given, but here the number of nematodes is important; after the experiment the nematodes were counted so the authors surely have this data. Whole species/nematode names i.e. with authors and date (Strongyloides papillosus, Haemonchus contortus) are already given in the introduction, so repeating them in subsequent chapters is unnecessary. When first time (abstract and main text) a species is mentioned in the text, its whole name should be given, that is, the common (English) name, the scientific name with the author(s) and the date. By the way, please complete the other species names (author/authors, date) used in the text, e.g. Meloidogyne javanica, etc, etc.

Done

A 0.001% solution of was also used ?, see Tables 2-5.

Thank you, we have indicated the concentration in Materials and Methods.

Tables 2-5: The notation "n=5" is inaccurate; this is, after all, the number of repeatitions, not the number of nematodes.

Done

Table 5:

The authors mention "rats" and "mice" - but specifically which species are involved. The common names are very vague - it is unclear what species were used in the experiment; it needs to be clarified/specified whether it is the burnished enggano rat Rattus adustus Sody, 1940, or perhaps the Australian swamp rat Rattus lutreolus (J. E. Gray, 1841) or the brown rat/ Norway rat Rattus norvegicus (Berkenhout, 1769), or the house rat/black rat Rattus rattus (Linnaeus, 1758). Using only common names is very confusing, especially since some species (e.g., rats) have several common names each, such as the aforementioned R. norvegicus and R. rattus. Unfortunately, it can happen that in the cited articles/databases this (species names) is not specified, and this is a serious mistake; often in veterinary works such errors are very common. So, in addition to the common names, you need to add scientific names (genera or species) to the title of this table. The comment also applies to the names listed in the Introduction section - mice, rats, cats, ets.

In the title to Table 5, we specified the genera of rats and mice: Rattus Fischer, 1803, Mus Linnaeus, 1766.

Please explain the abbreviations LC and LD in the legend

Done

Discusion and colnculison: These enumerations of compounds are very tiresome; they are repetitions; I doubt whether there is any point in listing them in the Discussion; in any case, they certainly need to be deleted from the Conclusion chapter.

Done

Reviewer 2 Report

The manuscript “Survival of nematode larvae Strongyloides papillosus and Haemonchus contortus under the influence of various groups of organic compounds” which I recently received for review, is an extensive and detailed paper authored by Boyko and Brygadyrenko. This is another work by these authors examining the effect of various compounds on the survival or activity of nematodes.

Introduction is written in the right form, materials and methods section consist all expected information, huge description of compounds used in laboratory diagnosis.

A large number of organic compounds used in the study (62) applied in 5 different concentrations allows for a thorough analysis of the effect of the tested compounds on the mortality of larvae of selected nematode species. Also, the results collected in 3 huge tables comprehensively present the effect of individual compounds (acyclic organic compounds, cyclic organic compounds and sulfur- and nitrogen- containing organic compounds) on the mortality of larvae and show the LC50 parameter.

The references list is extensive enough and contains up-to-date items.

I was just wondering why the latest version of Statistica was not used for statistical analysis. However, this is not an obstacle and is sufficient to calculate mean value and standard deviation.

Although the manuscript sent to me did not include line numbering, I do not find editorial flaws to be corrected and I consider the paper suitable for publication in this form.

Author Response

Dear reviewer,

Thank you for a detailed analysis of our paper.

I was just wondering why the latest version of Statistica was not used for statistical analysis. However, this is not an obstacle and is sufficient to calculate mean value and standard deviation.

The only licensed version of the Statistica we currently have is 8.0 (StatSoftInc., USA). Unfortunately, we are yet unable to purchase the latest version due lack of resources against the background of the war. Nonetheless, in the future when the opportunity arises we will certainly do our best to update the software.

Thank you once again for your work with our article and understanding!

Reviewer 3 Report

Major revision

The paper has high applied value for veterinary, authors did a lot of  hard work (!!), however some methodical and fundamental weakness points.

-        The most of the Discussion part can be transferred to the introduction because it not the discussion of the obtained results, but the information about chemicals and their use in another research, without distinct links with results of the paper.

-        What  kind of the Nematoda classifications was used by authors (it needs citation). They indicated two orders Rhabditida and Strongylida, however, both species belong to the order Rhabditida but to different families. Strongylidae and Trichostrongylidae (see the classification of De Ley & Blaxter, 2002, the modern classification for the Nematoda phylum, both studied species belong to Superfamily Strongyloidea within infraorder Rhabditomorpha, suborder Rhabditina, order Rhabditida).

-        Highly recommended to describe (and illustrate) the morphological criteria that the authors used to define the species, while they recognize the species using juveniles (J1,J2 and J3-invasive). Nematodes at the juvenile stage have poor morphology and scarce diagnostic characters, thus the species definition can be done only using the adult males in females in the host guts; however, this will be not the direct identification and it is highly recommended to use molecular technique for the species identification.

-        It is highly desirable to use the PCR diagnostics for the species definition at larval stages, and not only the morphological technique which is evidently not enough for the modern science.

-          Please describe the morphological characters used to define the juvenile stages: J1,J2 and J3-invasive. Do you use the juvenile mix or separate different larval stages – please tell distinctly.

- The hosts from which the nematode material was obtained characterized as ‘we used feces of caprines’. Please give the Table with all host Latin names, GIS of sampling, Latin names of helminths for every sample (one species or species mixture).

- Why autors used only mobile/immobile criterium to rtecognize the dead and live individuals and they do not use simple staining of individuals? it much more easy and distinct.

Author Response

Dear Reviewer,

Thank you for your work with our article and its thorough analysis. We have tried to make all the corrections you had suggested.

  1. The most of the Discussion part can be transferred to the introduction because it not the discussion of the obtained results, but the information about chemicals and their use in another research, without distinct links with results of the paper.

Done

  1. What  kind of the Nematoda classifications was used by authors (it needs citation). They indicated two orders Rhabditida and Strongylida, however, both species belong to the order Rhabditida but to different families. Strongylidae and Trichostrongylidae (see the classification of De Ley & Blaxter, 2002, the modern classification for the Nematodaphylum, both studied species belong to Superfamily Strongyloidea within infraorder Rhabditomorpha, suborder Rhabditina, order Rhabditida).

We had given the classification according to the FaunaEuropaea database that designates Haemonchus contortus, Strongylida order: https://fauna-eu.org/cdm_dataportal/taxon/84129b05-9cf3-4a1a-8f8b-6ade1ad40210. However, thanks to your suggestions, we made the changes according to the updated classification.  

  1. Highly recommended to describe (and illustrate) the morphological criteria that the authors used to define the species, while they recognize the species using juveniles (J1,J2 and J3-invasive). Nematodes at the juvenile stage have poor morphology and scarce diagnostic characters, thus the species definition can be done only using the adult males in females in the host guts; however, this will be not the direct identification and it is highly recommended to use molecular technique for the species identification. It is highly desirable to use the PCR diagnostics for the species definition at larval stages, and not only the morphological technique which is evidently not enough for the modern science.

You are absolutely right – the species are best identified using PCR diagnostics. Unfortunately, our laboratory is unable to carry out such analyses in the current circumstances (war in Ukraine). Therefore, we used the morphological characteristics according to VanWyk, A.; Cabaret, J.; Michael, L.M. (2004), and also VanWyk, J.A.; Mayhew, E. (2013). At the same time, in our previous studies (Boyko, O.O., Gugosyan, Y. A., Shendryk, L. I., &Brygadyrenko, V. V. (2019) Intraspecific morphological variation in free-living stages of Strongyloides papillosus (Nematoda, Strongyloididae) parasitizing various mammal species. Vestnik Zoologii, 53(4), 313–324. https://doi.org/10.2478/vzoo-2019-0030; Gugosyan, Y. A., Boyko, O.O., & Brygadyrenko, V. V. (2019). Morphological variation of four species of Strongyloides (Nematoda, Rhabditida) parasitising various mammal species. Biosystems Diversity, 27(1), 85–98. https://doi.org/10.15421/011913), we have provided an in-depth description of the morphological peculiarities that we used in our experiment. Nonetheless, we are grateful to you for pointing this out, since science keeps advancing, and therefore in the future we intend to seek for ways to resolve this question.

  1. Please describe the morphological characters used to define the juvenile stages: J1,J2 and J3-invasive. Do you use the juvenile mix or separate different larval stages – please tell distinctly.

In the experiment, we used a mixture of different-age larvae of Strongyloides papillosus, and separate studies were carried out on third-age Haemonchus contortus larvae. When we were identifying the species, we took into account body length, total maximum body width, length of tail end, length of the esophagus, and also specifics of its structure (filiform or rhabditiform with the bulbus), length of the intestine, and specifics of its structure as well (presence or absence of notable intestinal cells, their number, form, arrangement).

  1. The hosts from which the nematode material was obtained characterized as ‘we used feces of caprines’. Please give the Table with all host Latin names, GIS of sampling, Latin names of helminths for every sample (one species or species mixture).

In the experiment, we used feces of goat Capraae gagrushircus (Linnaeus, 1758) naturally infected by S. papillosus and H.contortus. Coordinates: 48.421341 N, 35.051363 E.

  1. Why authors used only mobile/immobile criterium to recognize the dead and live individuals and they do not use simple staining of individuals? it much more easy and distinct.

Thank you for the remark. Currently, there are various methods of determining vitality of nematode larvae. Further, we shall use your advice regarding staining. In this experiment, however, we counted dead and live specimens according to mobility and intestine deformation.

Round 2

Reviewer 1 Report

Two small comments:

1. In some places there is text written in Cyrillic (title of Table 2; legend of Table 3; first sentence of the Conclusions section) - please remember to remove this (now or during the author's proofreading).

2. Keywords: It seems to me that first should be "Strongyloididae" and then "Trichostrongylidae".

Reviewer 3 Report

Authors updated the ms using the best way in their conditions, thus I reccommend to accept  the paper.